



# Validation of aeroelastic dynamic model of Active Trailing Edge Flap system tested on a 4.3 MW wind turbine

Andrea Gamberini[1,2], Thanasis Barlas[2], Alejandro Gomez Gonzalez[1], and Helge Aagaard Madsen[2]

[1]Siemens Gamesa Renewable Energy A/S, Brande, Denmark
[2]DTU, Dept. of Wind and Energy Systems, Roskilde, Denmark

**Correspondence:** Andrea Gamberini (andgam@dtu.dk)

**Abstract.** Active Trailing Edge Flap (ATEF) is a promising technology for Wind Turbine load reduction and AEP improvement. However, this technology still needs extensive field validations to prove the reliability of the ATEF aeroelastic modeling codes. This article describes the validation of the dynamic response of the ATEF aeroelastic models developed for the BEM-based solvers HAWC2 and BHawC. The validation relied on field data from a 4.3 MW Wind Turbine (WT) equipped with an ATEFS on one blade and operating in normal power production. The validation consisted of three phases. At first, video recording of the ATEF deflection during WT operation allowed the tuning of the flap actuator model. In the second phase, the aerodynamic flap model was tuned and validated through the lift coefficient (Cl) transients measured with an innovative autonomous add-on measurement system placed on the blade in the middle of the spanwise extension of the ATEF. Finally, the aeroelastic ATEF model was validated based on the blade root moment (BMrM) transients over three months, from October to December 2020, with varying weather conditions. The validations showed that the simulations transient of Cl and MBrM are in good agreement with the corresponding measured transients, with a maximum difference for the blade-to-blade MBrM transients below 1% of the mean blade load during flap activation and below 1.7% during flap deactivation. An analysis of the possible root causes of these differences suggested additional measurements to improve the ATEF model tuning. The validation confirmed that the aeroelastic ATEF models provide a reliable and precise estimation of the impact of the flap on the wind turbine during flap actuation.

## 1 Introduction

In recent years, the steady growth in the size of utility-scale Wind Turbines (WT) led by the pursuit of lower levelized cost of energy resulted in a significant increase in the load carried by the WT components. One of the most promising technologies to mitigate the load increase consists of actively controlled flaps located at the blade trailing edge, the so-called Active Trailing Edge Flap (ATEF). From the pioneering works of Van Wingerden et al. (2008), Andersen (2010), Lackner and Van Kuik (2010),Aagaard Madsen et al. (2010) and Castaignet et al. (2011), to some of the more recent research by Bergami and Poulsen (2015), Barlas et al. (2016), Fischer and Aagaard Madsen (2016) and Bernhammer et al. (2016), several studies support that the integration of the ATEF in the WT design has the potential to reduce extreme and fatigue loads. These load reductions can be exploited to lower the components' cost or to increase the AEP, as shown by Pettas et al. (2016) and Abbas et al. (2023).



Despite the consensus on the potential benefits of active flaps in load reduction, the full-scale validation of ATEF aeroelastic engineering models and their potential load reduction is limited. Validating these models is challenging. An extensive code-to-code comparison of various existing models for simulating active flaps on rotating blades was performed in Prospathopoulos et al. (2021), where state-of-the-art Blade Element Momentum (BEM) models (hGAST, HAWC2, and FAST) were compared with higher fidelity models, including free-wake lifting line (GENUVP) and fully resolved Computational Fluid Dynamics

(CFD) models (MaPFlow and FLOWer). The comparison concluded that the BEM models cannot reproduce the correct distribution of the local thrust forces in the proximity of the flap edges because they neglect the 3D effects originated by the vorticity trailed from the edges and along the span of the flap section. However, these BEM models reasonably estimate the impact of an oscillating flap on the integrated overall thrust when the oscillating frequency is 1P. This result is explained by the over-prediction of the flap impact in the flap region being overall compensated by the flap impact under-prediction in the

blade regions near the flap edges. With the increase of the flap activation frequency, BEM model accuracy decreases. Finally, the study showed that modifying the BEM models to account for 3D effects due to the vorticity trailed from the flap edges (HAWC2 with Near Wake model and modified FAST) improved their prediction of both the local and the global impact of the flap on the thrust force.

ATEF subsystem validations have also been conducted through wind tunnel tests (e.g. Barlas et al. (2013)) and outdoor rotating

rig experiments Gonzalez et al. (2020). Recently, 3-dimensional lab-scale tests were performed within the large wind tunnel of the TU Berlin on BeRT, a 3 m diameter research turbine equipped with ATEF on each blade. The ability of different controllers employing trailing edge flaps to reduce fatigue (Bartholomay et al. (2022)) and extreme (Bartholomay et al. (2023)) flapwise blade root bending moments were assessed meanwhile providing datasets for future validation of numerical models. Regarding full-scale validation, only three field tests have been reported so far. These tests include the Sandia field test on a Micon 65/13

turbine (115 kW) Berg et al. (2013), the DTU and Vestas test on a V27 (225 kW) turbine Castaignet et al. (2014), and the Siemens Gamesa Renewable Energy (SGRE) and DTU tests on an SWT-4.0-130 (4.0 MW) turbine as part of the INDUFLAP2 project Gonzalez et al. (2020). Although these field tests confirmed the potential of active flaps in controlling aerodynamic loads, they also highlighted the need for further development and validation of numerical models for ATEF.

To address this gap, the Validation of Industrial Aerodynamic Active Add-ons (VIAs) project was carried out between 2019

and 2022 by SGRE and DTU, as described in Gomez Gonzalez et al. (2022). As part of this project, a prototype wind turbine with a rated power of 4.3 MW and a diameter of 120m (PT) was equipped with an pneumatically activated ATEF on a single blade. From May 2020 to February 2021, extensive testing of the active flap was conducted with different actuation strategies and flap deflection angles. The data collected in VIAs field tests allowed Gamberini et al. (2022) to validate for flap stationary activation state the ATEF model of the aeroelastic engineering tools BHawC (Fisker Skjoldan (2011)) and HAWC2 (Larsen

and Hansen (2023) and Aagaard Madsen et al. (2020)). The study relied on the measurements of the 10 minutes mean and maximum blade bending moments at the root of the three blades of the PT collected with the flap locked in a fully activated or deactivated position. A one-to-one validation approach was followed, where the aeroelastic simulations were performed under the wind conditions measured during the test campaign. The validation showed that the BHawC and HAWC2 tools equipped with ATEF agree with each other (difference within 2% for max and mean blade loads) and can estimate the blade loads with





accuracy within ±5% for ATEF stationary activation state.

The subsequent step in the ATEF model validation is the comparison of the aeroelastic response to the step actuation of the flap. In this article, the transient response of the BHawC and HAWC2 ATEF models during flap activation and deactivation is investigated and compared to the field data gathered in the VIAs project. The purpose of this validation is to enable reliable aeroelastic modeling of the load reduction strategies based on the actuation of trailing edge flaps, a fundamental milestone in

the design of future WT equipped with ATEF.

In this article, section 2 resumes the active flap system installed on the PT, and section 3 describes the PT aeroelastic model developed in BHawC and HAWC2 together with the structure of the ATEF model. The validation of the aeroelastic model is conducted in steps. The first step is tuning the actuator model, described in section 4. This step covers both the pneumatic system and flap sub-system response. Then the aerodynamic model is initially validated with a 3 hours field campaign where

the lift coefficient (Cl) was measured on a specific section of the PT, described in section 5. Finally, section 6 describes how the model is validated with a three months field campaign, covering a broad range of wind conditions but relying only on blade root loads. The overall results are then discussed in section 8.

## 2    Flap system and measurement setup

The VIAs project implemented an ATEF system on a single blade of the SGRE PT located at the Høvsøre test site in the

northwest of Denmark. Gomez Gonzalez et al. (2022) provides a detailed description of the active flap system, which consisted of a pneumatic supply system located in the hub connected with a hose to the active flap. The flap, shown on the left of Figure 1, was placed on the trailing edge of the outer 20 meters of the blade, between 64% and 98% of the blade radius. The pneumatic supply system had a remotely programmable control that regulated the pressure in the hose via a pressure valve. The air pressure in the hose controlled the flap movement; the higher the pressure, the higher the flap deflection amplitude.

Throughout the field campaign, the PT was equipped with a Data Acquisition System (DAS) system that continuously logged operational parameters, such as power, pitch, and rotor speed, with a $25\,Hz$ sampling rate. The same sampling rate was used to measure the flapwise and edgewise bending moments of all three blades by strain gauges located in the blades at $3\,m$ from the root. Also, the pressure at the pressure valve in the hub was recorded and integrated into the DAS. Furthermore, a met mast located approximately 2.5 D ($300\,m$) in front of the WT provided the wind speed and direction at three different heights, the

atmospheric pressure, the temperature, and the humidity.

In June 2020, an inflow and pressure measurement system was temporarily added to the PT to measure the aerodynamic properties at a specific ATEF section, as described in Madsen et al. (2022) and showed in the central photo of Figure 1. The system, developed by DTU, consisted of an inflow 5-hole Pitot tube sensor, a belt with 15 pressure taps, and an autonomous data acquisition and transmission system (flyboard). Both Pitot tube and flyboard were installed on the blade leading edge at

$50\,m$ from the hub flange, in the middle of the spanwise extension of the ATEF. The pressure belt was wrapped around the blade at a spanwise position of $49\,m$. The inflow and pressure measurement system provided data with a $100\,Hz$ sampling rate that was synchronized with the PT measurement data by the recorded GPS time. A close view of the pressure belt and flyboard



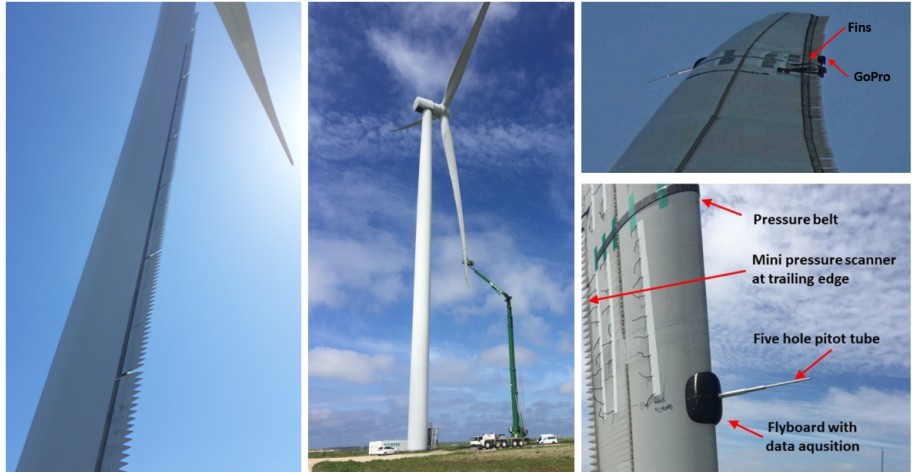

**Figure 1.** Left photo: Active flap placed on the trailing edge of the blade of the SWT-DD-120 turbine at the Høvsøre test field. Central photo: installation of the flyboard and pressure belt in June 2020. Bottom right photo: close look at the flyboard and the pressure belt. Top right photo: Camera and fins installed to measure the flap deflection. Photos curtesy of SGRE and DTU

is shown in the bottom right photo of Figure 1.

Additionally, two couples of small plastic fins were installed near the pressure belt location to analyze the ATEF deflection
visually. The fins of each couple lay aligned on the same blade section, with the two couples spaced around $0.5\,\mathrm{m}$ apart spanwise. For each couple, one fin was attached to the moving ATEF and the other to the blade structure. A GoPro camera was installed closely to record the flap deflection using the fins as reference points, as shown in the top right picture of Figure 1.

## 3   Aeroelastic setup

### 3.1   BHawC model

SGRE has internally developed and validated the aeroelastic engineering tool BHawC, based on the Blade Element Momentum (BEM). SGRE has furthermore developed a ATEF Module to model the flap aerodynamic and actuator system.

The BHawC model adopted in this paper was provided by SGRE and fine-tuned by the authors in Gamberini et al. (2022) where they showed the aeroelastic simulations were able to estimate the PT operational parameters with an accuracy of $\pm3\%$ and blade loads within $\pm2\%$ for the condition of stationary flap state. The current paper covers the tuning and validation of the
ATEF Module actuator and the part of the aerodynamic model responsible for the aeroelastic response to the flap actuation.

### 3.2   HAWC2 model

HAWC2 is the aeroelastic engineering tool developed by DTU Wind. It is based on BEM and models the unsteady aerody-
namics associated with the active flaps with the Beddoes-Leishman type ATEFlap dynamic stall model Bergami and Gaunaa



(2012). The authors tuned the HAWC2 model of the SGRE WT prototype in Gamberini et al. (2022), based on the BHawC
model, obtaining negligible code-to-code differences (max difference below 1%) for mass properties and WT operational parameters and a max difference within 4% for the blade loads. For the ATEFlap model, the suggested values of the coefficients for the indicial response exponential function and the exponential potential flow step response were used. These parameters were tuned to describe the step response of the NACA 64-418 profile (18% thickness) that can be considered an acceptable approximation for the modeled ATEF.

### 3.3 Modeling of the Flap system

The ATEF system installed on the PT was simplified as a controlled pneumatic system that regulated the pressure inside a hose connected to the flap located on the trailing edge of one blade. Increasing the air pressure inflated the hose that deflected the flap. The flap deflection changed the blade profile shape, consequently modifying the local aerodynamic forces and affecting the loading of the whole PT.

In the ATEF aeroelastic model, depicted in Figure 3, the pneumatic system and the flap structure were merged in the actuator model. This model linked the controller signal directly to the flap deflection, disregarding the air pressure signal. This simplification was possible because the pneumatic system did control only the final pressure value but did not control the pressure transient. This transient depended only on the pneumatic system layout. Therefore the air pressure and the consequent flap deflection were expected to have a constant transient for every defined activation pressure value.

The actuator model provided the flap deflection to the aerodynamic model that computed the dynamic aerodynamic properties of the flap section, needed by the aeroelastic code to compute the PT loads. In both codes, the ATEF aerodynamic model relied on the stationary lift and drag coefficient (Cl and Cd) curves of the flap profiles for both active and not active flap states. SGRE provided the aerodynamic characteristics of the 21% thickness flap profile. Similarly to the method described in Gomez Gonzalez et al. (2018), SGRE measured the aerodynamic characteristics in a wind tunnel campaign performed at the Low-Speed Low-Turbulence wind tunnel facilities of the faculty of aerospace of TU Delft. The measurements, run at a Reynolds number of mainly 4 million, focused on the 21% thickness profile where three different shapes of the deflected flap were modeled with a corresponding fixed add-on. The measurements provided the aerodynamic characteristics for the flap that was not active, which was activated with low pressure and activated with high pressure. The data for middle activation pressure were instead obtained by interpolation, as previous tests in the VIAs project showed a linear relation between flap deflection and pressure for the studied ATEF system. To fully simulate the flap on the PT, the aerodynamic characteristics of the 24% and 18% thickness flap profiles were needed. They were computed assuming that the same flap deflection leads to the same lift and drag variation across the family of flap profiles. Under this assumption, from the 21% thickness profile, the lift and drag increases due to a specific flap deflection were calculated as a function of the angle of attack. These ΔCl and ΔCd were added to the 24% and 18% thickness profile properties after being linearly scaled to adjust to the actual chord length of the new profiles. Figure 2 shows an example of the normalized Cl curves for flap not active (black line) and flap active at low pressure (red line) of the 21% (left), 18% (top right) and 24% (bottom right) thickness profiles. It also shows the ΔCl obtained from the 21% thickness profile (center). The ATEFlap model of HAWC2 required the aerodynamic properties to be in a specific format, where aero-



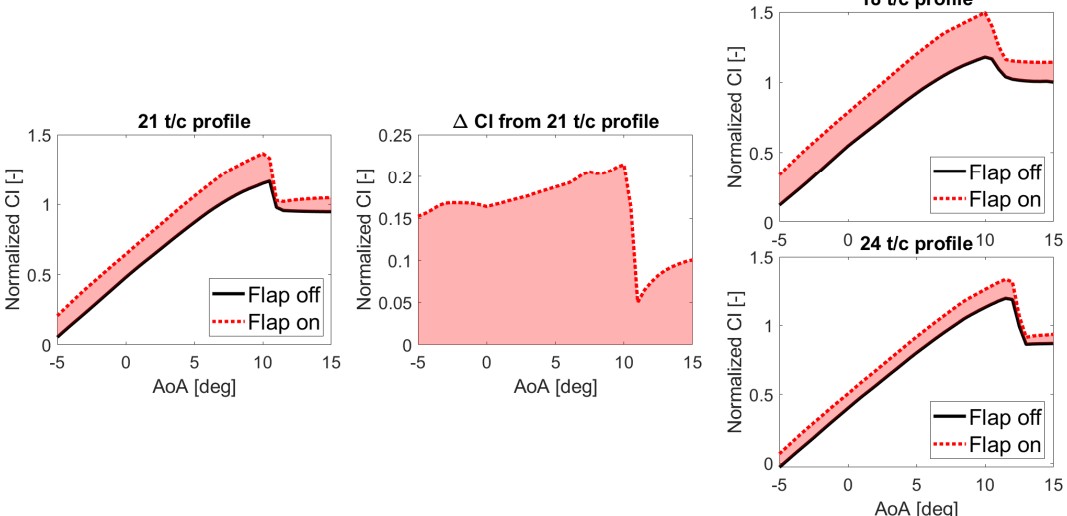

**Figure 2.** Left: Normalized Cl curve of 21% thickness profile with flap not active (black) and flap active at low pressure (red dotted line). Center: ΔCl from 21% thickness profile due to flap activation. Right: Normalized Cl curve of 18% (Top) and 24% (Bottom) thickness profile with flap active at low pressure (red dotted line) obtained by adding the scaled ΔCl from 21% thickness to the curves with flap not active (black lines).

dynamic coefficients were given as a function of the angle of attack and the angle of flap deflection. This format was obtained utilizing the Preproc ATEFlap tool. BHawC and HAWC2 aerodynamic models assume a linear proportional relationship be-

tween the flap deflection and the variation of the static Cl and Cd. Both codes derive the dynamic properties with a partially different approach, mainly due to different dynamic stall models, properties that are then passed to the global WT aeroelastic model.

The tuning and validation of the ATEF model was performed in three steps. At first, the actuator model was tuned to match the actual flap deflections obtained on the PT for the required activation flap pressure. The second step consisted of an initial

validation based on the Cl measurements obtained from the flyboard installed at the PT flap location in a three hours long field campaign. Finally, the ATEF aerodynamic model was validated based on blade root load transients measured in a three months long field test.

# 4  Actuator model tuning

To validate the aerodynamic model, a reliable flap actuator model was needed to compute the flap position and its transient. In

this paper, the development and tuning of the flap actuator model were obtained by the analyses of video recordings of the PT flap actuation.

In June 2020, a GoPro camera and two sets of plastic fins were temporarily installed on the PT blade. The camera captured





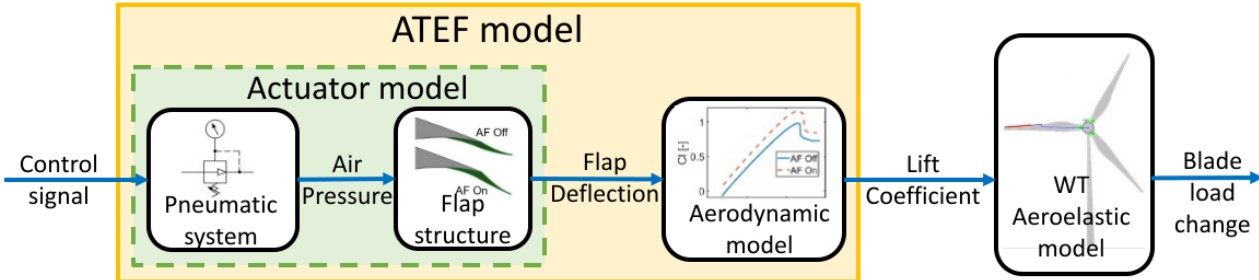

**Figure 3.** Structure of the ATEF aeroelastic model implemented in BHawC and HAWC2

the movements of the flap during its activation and deactivation under three distinct activation pressure levels (low, middle, and high) and two operational states of the wind turbine: idling mode and normal operation at 6 rpm. Each combination was
repeated four times to ensure data reliability. The video analysis and modeling tool Tracker (Brown et al. (2023)) was utilized to extract the flap deflection in each video by tracking the relative position of the two fins during the flap activation and de-activation. For each combination, the four recorded cases consistently yielded a similar flap deflection transient during flap activation; their binning and normalization resulted in the characteristic flap activation deflection transient (activation deflec-tion curve) that ranged between 0 (flap not active) and 1 (flap fully deployed). The characteristic flap deflection transient for
deactivation (deactivation deflection curve) was obtained with a similar process.

The comparison of the flap deflection transients showed a substantial overlap of the curves obtained for middle and high acti-vation pressure, with the low-pressure case being slightly faster. Although increasing the activation pressure resulted in higher flap deflection, the normalized flap deflection transient remained independent of pressure from middle activation pressure and above. Furthermore, the flap displayed a slower activation and a faster deactivation when the WT was in normal operation
compared to the idling state. The latter behavior can be attributed to the effect of the aerodynamic pressure distribution on the blade section, which generates an aerodynamic force that opposes the flap deflection during activation and supports it during deactivation. The magnitude of this aerodynamic force, higher in the normal operation state compared to the idling state, di-rectly affects the time required for full activation and deactivation, with higher forces resulting in slower activation and quicker deactivation.

In modeling the flap actuator, we assumed the activation and deactivation curves were independent of the actual activation pressure. This assumption was valid for high and middle activation pressure scenarios which are most of the available PT field data. The actuator model should also include the impact of the aerodynamic loads on the flap dynamic, which varies in function of the wind speed and WT operational state. Therefore, we selected the middle and high-pressure deflection transients for normal production, assuming a negligible impact of the aerodynamic load change around the measured operative condition.
The impact of the aerodynamic loads on the total flap deflection was also neglected based on the results from Gamberini et al. (2022), where the stationary flap properties were validated for a wide range of wind speeds.

On the PT pneumatic system, the signal of the flap controller was not recorded. Therefore the pressure channel was used to



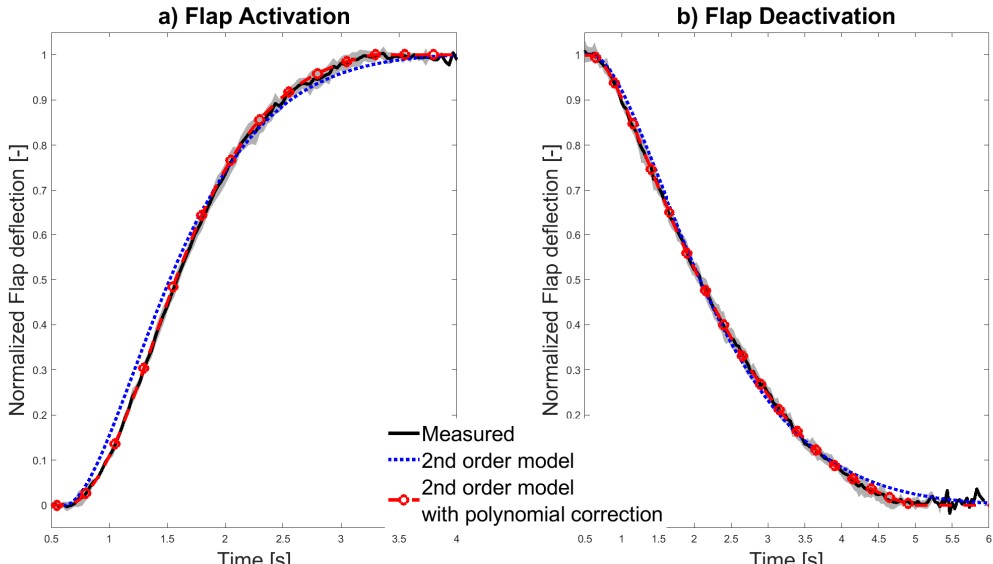

**Figure 4.** Normalized flap deflection transients measured with the video tracking (black line) from the second-order transfer function model (blue dotted line), and the second-order transfer function model fine-tuned with a fourth-order polynomial function (red dashed line with circles) for flap activation (figure a) and deactivation (figure b). The measured lines are shown with the mean value line enveloped by the grey 1 std error band

initially identify the controller activation time, assuming no delay between the controller activation and the opening of the pressure valve. The flap actuator was finally modeled as a simple second-order transfer function without poles. A fourth-order

polynomial function was added to improve the similarity with the deflection curve data. In Figure 4a, the activation deflection curve for middle and high pressure (black line) with the error band of 1 standard deviation (std) is compared with the modeled second-order transfer function (blue dotted line), and the improved model with the fourth-order polynomial function (red dashed line with circles). Similarly, Figure 4b compares the experimental and modeled flap deactivation transients.

## 5    ATEF Aerodynamic model validation on a WT blade section

The aerodynamic model of the ATEF aeroelastic module was initially validated for a single PT blade section equipped with the active flap. This validation relied on a three hours measurement data set focused on one wind speed. It compared the transient of the lift coefficient measured on the PT blade with the Cl transient calculated via aeroelastic simulations during the flap activation and deactivation. The validation furthermore included the comparison of the measured and simulated transient of the blade-to-blade flapwise bending moment at the root of the blades.



## 5.1 Measurements

In June 2020, an inflow and pressure measurement system was temporarily mounted on the PT in the middle of the blade span equipped with the active flap for around one day (Madsen et al. (2022)). The measurement system comprised a pressure belt with 15 taps, a 'flyboard' with the data acquisition system, and a five-hole Pitot tube measuring the local inflow to the blade section. The data acquisition system in the flyboard sampled the data from the pressure scanners and the five-hole Pitot tube with a sampling rate of 100 Hz. The captured raw data from the flyboard and pressure belt were processed and converted into quantities of interest. This process includes the calibration of the Pitot tube pressure data and conversion into two flow angles (inflow angle and sideslip angle) and a velocity. The inflow measurement quantities were further corrected to transform the Pitot tube inflow angle into the airfoil angle of attack AoA and relative velocity. This transformation was based on 2D CFD simulations performed by SGRE on the airfoil section for zero and full flap states, where the flow velocity and angle at the point of measurement of the Pitot tube were extracted as a function of the geometric angle of attack (AoA), see Madsen et al. (2022). The pressure belt data was integrated into 2D aerodynamic forces by chordwise trapezoidal integration of the pressures, where a trailing edge pressure was added as an average of the two nearest points. The local pressure and lift coefficients (Cl) were calculated using the corrected dynamic pressure and angle of attack from the Pitot tube. Due to uncertainties in the conversion process based on 2D CFD, the absolute value of AoA, and the consequent absolute Cl, could not be used for the validation. Therefore, the lift coefficient validation focused on comparing the Cl transients unaffected by the conversion uncertainties. During most of the measurement period, the wind speed was relatively low, between 4 and $6\,\mathrm{m\,s^{-1}}$, keeping the wind turbine close to the minimum operational rotor speed. For the validation, a three-hour time interval characterized by a relatively constant wind (between 4 and $5\,\mathrm{m\,s^{-1}}$ 10 minutes mean wind speed) with low turbulence intensity (below 10%) was selected from the measurement period. The low variation of mean wind speed and low turbulence intensity were beneficial in reducing the variability range of the lift coefficient, facilitating the calculation of the average Cl transient curve during flap actuation. In this selected time interval, the flap was performing on-off actuation cycles, switching between $60\,\mathrm{s}$ at full middle-pressure activation and $60\,\mathrm{s}$ at complete deactivation, for a total of $90\,\mathrm{s}$ full activation and deactivation cycles.

## 5.2 Aeroelastic simulations

The same set of aeroelastic simulations was performed with BHawC and HAWC codes to compute the average Cl transient curve during flap activation and deactivation. The Cl was affected by the variation of the wind field along the rotor plane (due to wind shear or wind veer, for example) and by the rotor tilt and cone angles. The average Cl transient curve was obtained by averaging simulations with flap actuation occurring at different azimuth angles (FA angle), reducing the Cl azimuthal fluctuation's impact on the transient estimation. Precisely, the set consisted of 12 simulations with FA angles evenly spaced. All simulations were 2 minutes long and were run at a constant wind speed of $5\,\mathrm{m\,s^{-1}}$, without turbulence, at the standard air pressure, and with a wind shear of 0.2. The flap started in the deactivated position, was activated at t=30 s, and deactivated after 60 s, at t=90 s, reproducing the activation and deactivation cycles performed on the PT. The flap actuator model of both BHawC and HAWC PT aeroelastic models were updated to match the tuning described in Chapter 4. The Cl was calculated at the blade





section where the pressure measurement system was installed. To minimize the difference between the two aeroelastic models, the aerodynamic setup of the models was configured as similarly as possible. Both models had the BEM implemented on a
polar grid, as described in Aagaard Madsen et al. (2020), to better account for the rotor induction imbalance due to the flap equipped on only one blade. Furthermore, they implemented a potential flow tower shadow model linked to the tower top movements.

## 5.3  Average lift coefficient transient

The rotor cone and tilt angles of the wind turbine, along with the variations in wind speed resulting from wind shear, veer and
large turbulence structures, contribute to periodic fluctuations in the lift coefficient. These fluctuations challenged the detection and characterization of the transient Cl caused by flap actuation. The averaging of cases with flap actuation occurring at different azimuthal positions substantially reduced the azimuth-dependant variation of Cl, with the average Cl transient quality improved as the flap actuation azimuthal positions were evenly and balanced spread.

In the case of the aeroelastic simulations, the average Cl transient was derived through the binning based on the simulation time
of the 12 simulations with evenly spaced FA angles. This procedure was performed for both BHawC and HAWC2 simulations. Figure 5a shows the Cl time series (dotted lines), normalized to the average Cl increase, from the BHawC simulations around the flap activation time (t=0 s). The azimuth-dependant oscillations of Cl amount to around 40% of the Cl variation due to flap activation ($\Delta Cl\_F$) and affect the slope and shape of Cl transient during the flap activation. Instead, the average Cl transient (black line) is constant before the activation and rises at an almost constant rate until it settles to a constant value after approx-
imately 2.5 s. A similar pattern, but with a longer settling time of around 5 s, is exhibited by the averaged Cl transient during flap deactivation, as shown in Figure 5b. The results from the HAWC2 simulations were almost identical to the BHawC results and are not included in this paper for brevity.

Regarding the measurement data, the times of flap actuation (both activation and deactivation) were determined by identifying the instant at which the gradient of the flap actuation pressure started to change, as the flap valve was located directly at the
output of the supply valve. Subsequently, the Cl time series were segmented into temporal windows centered around the flap actuation time and synchronized accordingly. The average Cl transient was finally obtained by binning the Cl signals based on the time window. Different filtering techniques were tested to reduce the oscillation of the averaged Cl transient caused by turbulence and measurement noise. The best results were achieved with a low-pass, zero-phase digital filter set to attenuate frequencies above 9P (9 times the rotor rotational frequency) without affecting the slope of the Cl transient. For accurate
validation, synchronizing the Cl measurement with the PT measurements was crucial in ensuring the precise timing of the Cl variation relative to flap actuation pressure. Initially, the synchronization relied on the GPS time recorded by the flyboard, which was further refined by aligning the time of maximum acceleration along the blade length measured on the flyboard with the time the blade was oriented toward the ground. During the average Cl calculation, some measurements were discarded due to insufficient data quality, leading to 46 measured transients being used for activation and 63 for deactivation. The measure-
ments had an acceptable distribution of the flap actuation azimuthal angle, as shown in Table 1, with three cases or more for all sectors. Figure 5c shows the normalized measured Cl transient during flap activation (dotted lines). These transients exhibit





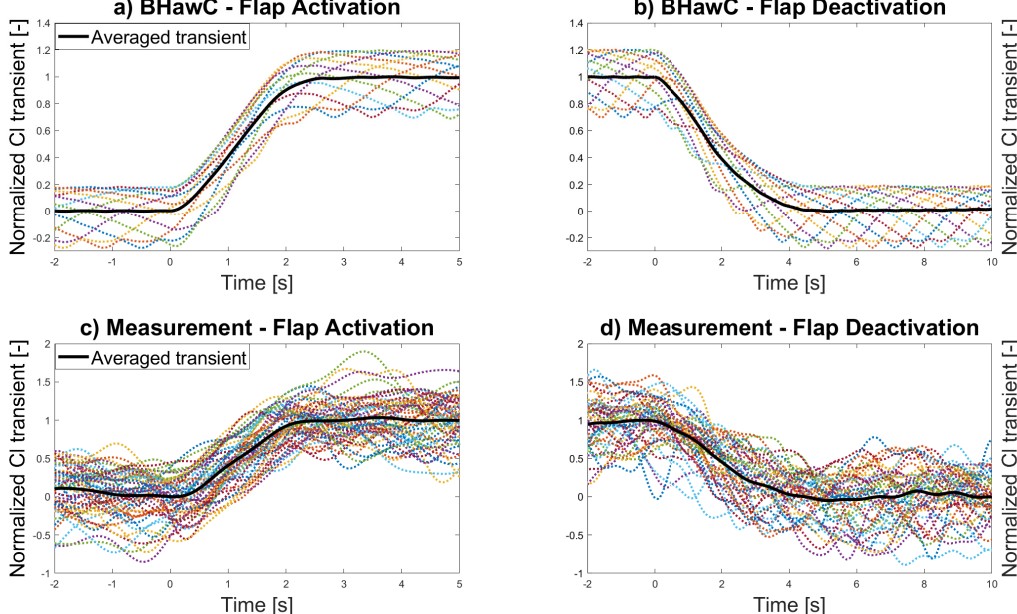

**Figure 5.** Figures a and b show the normalized averaged Cl transient (black line) for flap activation and flap deactivation, respectively, obtained from the averaging of the Cl transient of 12 BHawC simulations (dotted lines). Similarly, Figures c and d show the normalized averaged Cl transient (black line), respectively, for flap activation and deactivation, obtained from averaging the measured Cl transient (dotted lines).

a higher variability than the aeroelastic simulations, with a range of the same order of amplitude of the increase in Cl due to flap activation, variability mainly caused by the wind turbulence (omitted in the simulations due to the difficulties in estimating the correct value of turbulence for very short simulations). Nevertheless, the averaged Cl transient (black line) exhibits a clear and almost linear variation from a slightly decreasing value before the activation to an almost constant value after the activation. The averaged Cl transient during flap deactivation (black line in Figure 5d) shows a higher fluctuation behavior than the activation transient but is still considerably smoother than the measurements.

### 5.3.1 Blade-to-blade azimuth based blade root moment

Another crucial aspect for validating the aeroelastic model of the ATEF was the analysis of the blade root moment transient resulting from flap actuation. However, measuring this load transient proved challenging due to its high-frequency response, often hidden within the complex dynamics of the blades responding to factors such as turbulence, shear, vibrations, and rotation. Gomez Gonzalez et al. (2021) introduced a blade-to-blade (b2b) analysis method to compute the load transient caused by the ATEF actuation. This approach involved calculating the difference between the loads acting on the blade with the flap and the load acting on another blade without the flap, but delayed by a time corresponding to a third of one rotor rotation. This artificial





| | | | **Number of measured transients per flap actuation azimuth angle** | | | | | | | | | | |
|---|---|---|---|---|---|---|---|---|---|---|---|---|---|
| | *Total* | *0* | *30* | *60* | *90* | *120* | *150* | *180* | *210* | *240* | *270* | *300* | *330* |
| **Cl activation** | 53 | 4 | 7 | 3 | 3 | 3 | 2 | 5 | 5 | 5 | 2 | 9 | 5 |
| **Cl deactivation** | 46 | 3 | 3 | 3 | 3 | 4 | 3 | 4 | 6 | 4 | 5 | 4 | 4 |
| **MBr activation** | 69 | 6 | 8 | 4 | 4 | 4 | 9 | 5 | 5 | 6 | 3 | 9 | 6 |
| **MBr deactivation** | 69 | 7 | 5 | 3 | 9 | 5 | 5 | 4 | 5 | 6 | 8 | 6 | 6 |

**Table 1.** Number and azimuth distribution of measured cases used for the calculation of the average Cl transient and average blade to blade moment difference (MBr) during flap activation and deactivation.

time shift aimed to synchronize the load time series of blades in the same azimuthal position, thereby mitigating the influence of periodic signal dynamics resulting from rotation, forced vibrations, and wind shear.

     This paper proposes a novel azimuth-based b2b method (az-b2b) calculating the load difference by interpolating the loads based on the cumulative azimuth position instead of relying on time shifting. The method comprises three steps. Firstly, the Cumulative Sum of the azimuthal Angle (CSA) is calculated for each blade. Secondly, the load of the blades without the flap

is interpolated as a function of the CSA of the blade with the flap. Lastly, the difference between the load of the blade with the flap and the interpolated load of the blade without the flap is calculated.

     Notably, the az-b2b method eliminates the need to initially segment the time series around the relevant event, as the previous b2b method requires. It can be applied directly to the entire time series in a single run. Additionally, the az-b2b method is not dependent on the calculated mean rotor speed, which is influenced by the time extension of the segments. This characteristic

makes it less sensitive to minor variations in rotor speed. By ensuring that the load difference is between two blades positioned at the same azimuthal location, the az-b2b method effectively reduces azimuthal-dependent load fluctuations. However, precise measurement of the azimuthal angle is essential to avoid errors during the interpolation phase. Like the original b2b method, the az-b2b method is still sensitive to high rotor speed variation that can result in a significant variation of data density for azimuthal angle, potentially affecting the quality of the interpolation result.

The az-b2b method was utilized to calculate the differences in flapwise bending moments at the blade root (MBr1 and MBr2) between the blade with flap (BF) and the remaining blades (B1 and B2) in the aeroelastic simulation sets of BHawC and HAWC2. As shown later in this paper, the two blade-to-blade differences did not differ significantly in both simulations and measurements; therefore, the mean MBr (MBrM) of the two b2b differences was used as reference transient. Finally, the binning of the twelve simulations as a function of the simulation time computed the average MBrM transient. Figure 6a shows

the MBrM transients (dotted lines) obtained from the BHawC simulations, together with the average MBrM (black line). All the signals were normalized in order to make the increase of the b2b moment due to flap actuation unitary. The azimuthal variability of the MBrM transients is present but significantly lower than the Cl transients, being less than 10% of the average transient. The average MBrM transient is almost constant before the flap actuation, and it rises smoothly from around 0.5 s until it converges to an almost constant value after 2 s. The average transients of the individual b2b differences (MBr1 and





MBr1 represented by a dotted blue line and a dotted red line, respectively) exhibit only a small difference between themselves. Similar considerations are valid for the MBrM transients during flap deactivation shown in Figure 6b. The MBrM transients from the HAWC2 simulations were almost identical to the BHawC results and are not included in this paper for brevity. Regarding the measurement data, the same methodology employed for the Cl transient was applied to compute the blade

load difference. The measurement signals were segmented into temporal windows centered around the actuation time and

synchronized accordingly. Subsequently, the consistency of the azimuthal angle signal was verified, and the rotor speed was leveraged to compute the missing or erroneous values. Afterward, the MBrM transients were computed, and the average MBrM was obtained using the binning approach. In the average MBrM calculation, the same number of measured time series (69) were used for flap activation and flap deactivation. The distribution among the rotor sector of the flap actuation azimuthal angle was also acceptable, with most sectors having at least five cases. Similarly to the results from the Cl transients, the MBrM

measured transients (dotted lines in Figure 6c) show a high oscillation, mainly caused by the wind turbulence, with a range comparable with the load increase due to the flap activation. Nevertheless, the averaged MBrM transient (black line) is almost constant before the activation and smoothly increases and converges to the flap actuation value with minor oscillations. The average MBr1 and MBr2 are also close to each other, with a max difference of 0.1. Similar considerations can be done for the flap deactivation case, shown in 6d.

**5.4 Validation results and discussion**

In comparing the simulation results with the measurements, the actuation pressure signal was used to synchronize the transients. In detail, the time the flap pressure gradient undergoes a quick change was aligned with the flap activation time in the simulations. For the deactivation case, an additional delay of $0.3\,\mathrm{s}$ was added to the simulation flap deactivation time to properly align the Cl and blade-to-blade loads with the measurements. We believe this delay is related to the structure of the pneumatic

actuator system, and it would have been identified during the actuator system tuning if the controller signal had been available. For the activation phase, a time delay was not clearly needed.

The simulated and measured Cl transients are compared in Figures 7a for flap activation and 7b for flap deactivation. In the figures, the synchronized flap actuation time is indicated by the estimated flap command (black dashed line), used in the simulation to actuate the flap, and the measured flap pressure signal (blue dotted line). The transients from BHawC (blue dashed line with x marker) and HAWC2 (red dashed-dotted line) simulations closely match each other, with a maximum difference below

6% of the $\Delta$Cl generated by the flap actuation in both flap activation and deactivation cases, mainly caused by an offset of a single time step $(0.04\,\mathrm{s})$ between the two transients. The simulated Cl has transients significantly similar to the measurements (black line with round markers). In the activation case, the maximum difference is below 40% of the measurements std, below 8% of the $\Delta$Cl, with the simulated Cl starting to increase $0.1\,\mathrm{s}$ (5% of the time for full flap activation) before the measurement

Cl but at a lower slope, and it converges to full activation with a delay of $0.15\,\mathrm{s}$ (7.5% of the full flap activation time). In the deactivation, the maximum difference is below 6% of the measurements std, below 8% of the $\Delta$Cl, and the simulated Cl starts to decrease with a delay of $0.2\,\mathrm{s}$ (4% of the full flap deactivation time) that quickly recovers, and afterward it precedes the measurement slope of less than $0.1\,\mathrm{s}$ (2% of the full flap deactivation time) until full deactivation. The std of the simulated

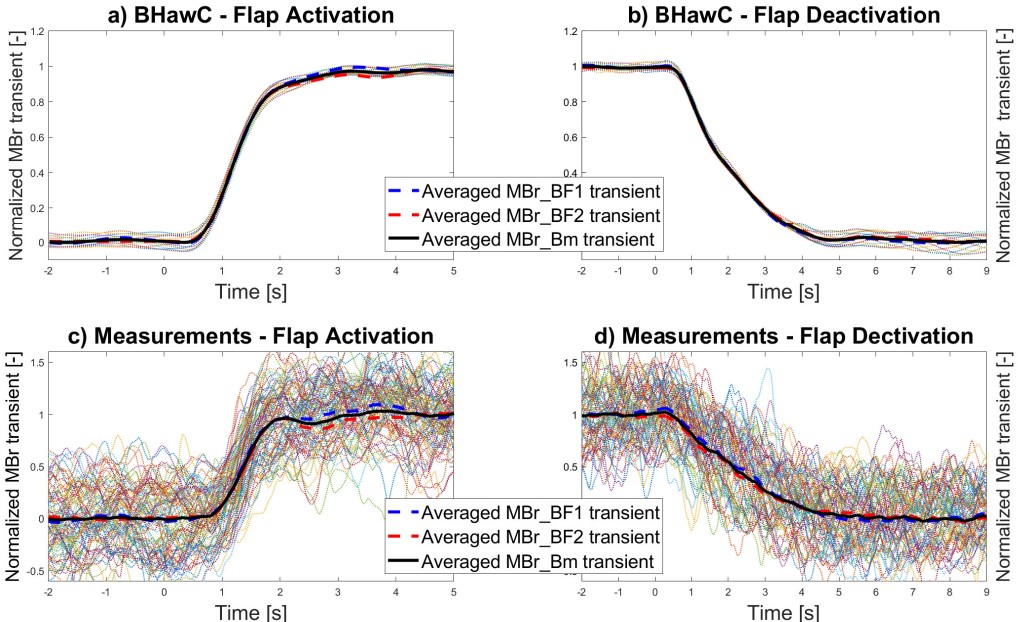

**Figure 6.** Figure a and b show the normalized averaged b2b bending moment difference at blade root (MBrM) transient (black line) for flap activation and flap deactivation respectively, obtained from the averaging of the MBrM transient of 12 BHawC simulations (dotted lines). Similarly Figure c and d show the averaged normalized MBrM transient (black line) for flap activation and deactivation respectively, obtained from the averaging of the measured MBrM transient. All the plots also include the b2b bending moment difference between the blade with flap and the blades without (MBr1: dashed blue line, MBr2: dashed red line)

transients are often equivalent, and nearly half of the measurement std. This difference is mainly due to the omission of turbu-

lence in the aeroelastic simulations.

The b2b moment transients are compared in Figures 8a for flap activation and 8b for flap deactivation. Similarly to the Cl comparison, the MBrM transients from BHawC (blue dashed line with x marker) and HAWC2 (red dashed-dotted line) simulations closely match each other, with a maximum difference below 7% of the $\Delta$MBrM due to the flap actuation in both flap activation and deactivation cases, mainly caused by an offset of $0.06\,\mathrm{s}$ between the two transients. Both transients have a std significantly

smaller compared to the Cl transients, showing the benefit of the az-b2b method in removing the impact of the azimuthal load oscillations. HAWC2 transient having a std almost twice the std of the BHawC simulations is mainly related to higher rotor speed oscillations caused by a not perfect implementation of the SGRE controller in the HAWC2 code. The simulated MBrM has transients similar to the measurements (black line with round markers). In the activation case, the maximum difference is below 55% of the measured std and 17% of the $\Delta$MBrM due to the flap activation. Mirroring the Cl transients, the simulated

MBrM begins to increase $0.2\,\mathrm{s}$ (10% of the full flap activation time) before the measured MBrM but at a lower slope, and it converges to full activation within the measurement fluctuation transient. In the deactivation, the maximum difference is below





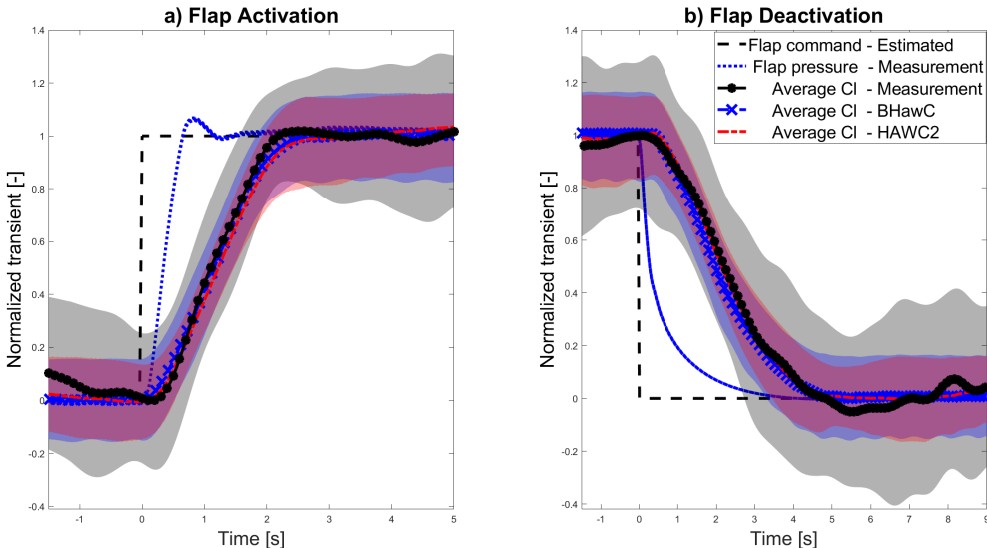

**Figure 7.** Comparison of the normalized averaged Cl transient obtained from BHawC (blue dashed line with x marker) and HAWC2 (red dash-dotted line) simulations and measurements (black line with asterisk marker) during flap activation (a) and deactivation (b), including an error band of 1 std of the matching colors. The measured flap pressure (blue dotted line) and the estimated flap command (black dashed line) are also included.

60% of the measured std, below 20% of the $\Delta$MBrM, and the simulated Cl starts to decrease with a delay of $0.2\,\text{s}$ (5% of the full flap deactivation time) that quickly recovers and afterward precedes the measurement slope of less than $0.1\,\text{s}$ until full deactivation.

Another purpose of the initial validation was tuning the ATEF model to ensure the proper synchronization between the simulated and measured Cl and MBrM transients. Figure 9a shows all the ATEF aeroelastic model signals relevant for the model tuning during flap activation. The measured controller signal was not recorded in the measurement campaign. Therefore, the measured flap pressure (blue dotted line) synchronizes the simulated flap control signal (black dotted line). This control signal commands the flap deflection (red dashed line with circular markers), deflection obtained from the post-processing of the flap

deflection videos (green squared markers). The Cl transient follows the flap deflection by a few milliseconds in the simulations (blue line with squares and orange line) and by $0.1\,\text{s}$ in the measurements (grey line with arrow marker). The slight difference between the simulated and measured Cl transients (especially if compared to the measurements std) confirms the current model( with the linear relation between flap deflection and Cl variation) provides reliable results. After the Cl increase, the MBrM transient starts $0.5\,\text{s}$ later, with the simulations anticipating the measurements of $0.2\,\text{s}$. Figure 9b shows the signals

relevant for the ATEF model tuning during flap deactivation. To properly align the simulations Cl and MBrM transients with the measurements, a delay of $0.3\,\text{s}$ was introduced in the actuator model between the flap controller and the flap deflection during deactivation only.





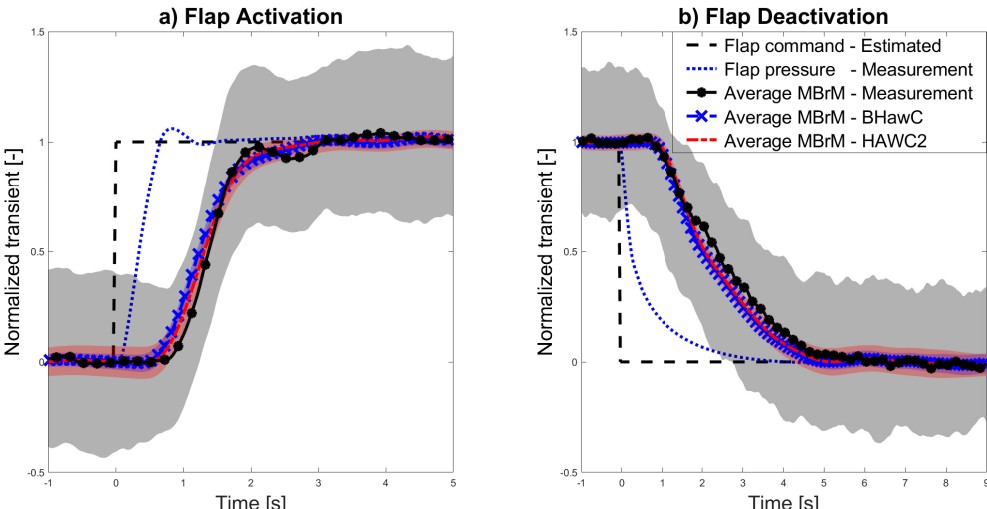

**Figure 8.** Comparison of the normalized averaged b2b blade root bending moment (MBrM) transient obtained from BHawC (blue dashed line with x marker) and HAWC2 (red dash-dotted line) simulations and measurements (black line with asterisk marker) during flap activation (a) and deactivation (b), including an error band of 1 std of the matching colors. The measured flap pressure (blue dotted line) and the estimated flap command (black dashed line) are also included.)

In conclusion, the initial validation showed a good agreement between the simulated and measured Cl and MBrM transients. However, small differences in the transients motivated the broader validation described in the following section.

## 6  Extended ATEF Aerodynamic model validation

The validation of the aerodynamic model of ATEF aeroelastic was extended to a wider range of environmental conditions with the measurements obtained with the PT field campaign between October and December 2020, when the ATEF system was tested for different pressure activation and activation patterns. In this test campaign, the flyboard was not present, and the validation did not rely on the Cl measurements but only upon comparing the transient of the blade-to-blade moment at the

blade root (MBrM) during flap activation and deactivation. The validation followed the so-called one-to-one (o2o) approach, where the measurements are compared with a set of simulations reproducing the WT operating under the same environmental conditions measured at the time of the flap actuation. As the validation focused on the short transient happening within $10\,\mathrm{s}$ after the flap activation and deactivation, the simulations did not rely on the 10 minutes averaged environmental condition but on the actual condition measured at the flap actuation time.






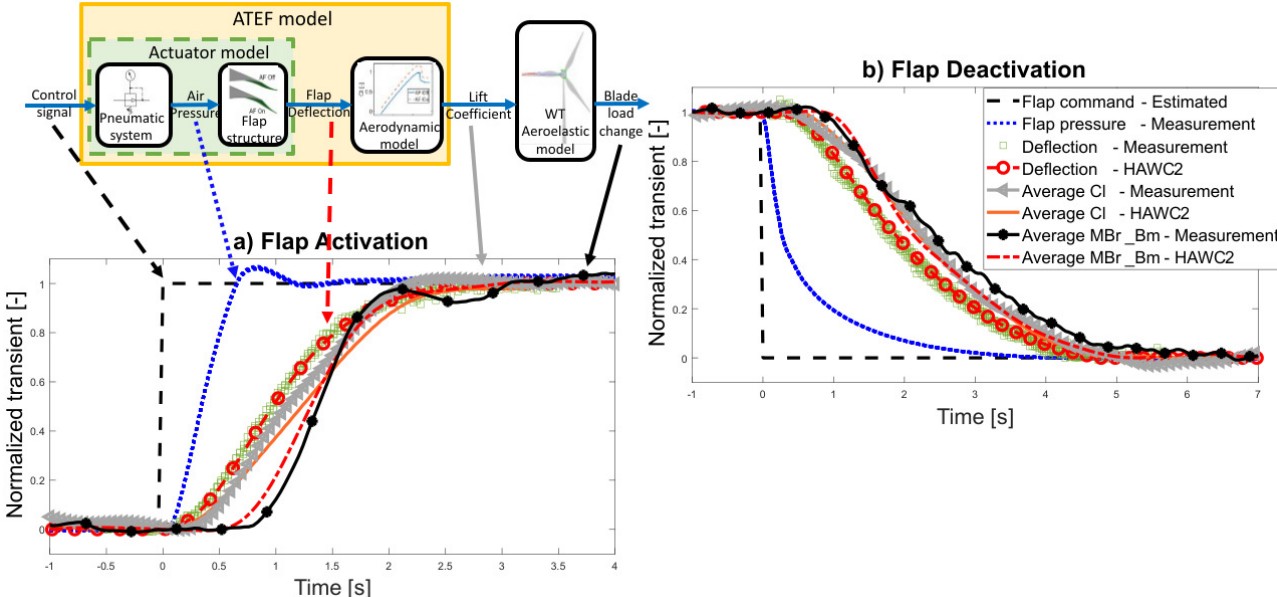

**Figure 9.** a) Signals relevant for the modeling and validation of the flap model during flap activation, linked to the ATEF model scheme. Transients of Flap controller command (black dashed line), flap pressure (blue dotted line), measured and estimated flap deflection (green squares and red dashed line with circles), measured and estimated Cl (gray line with triangle and red line) and measured and estimated MBrM (black line with asterisks and reddash-dotted line) are plotted to verify the time tuning of the aeroelastic model. b) Signals relevant for the modeling and validation of the flap model during flap deactivation

## 6.1 Field campaign

Between October and December 2020, the ATEF system on the PT was tested for several actuation pressure and activation pattern. The measurements of full activation and deactivation cycles at middle activation pressure with the WT operating in normal production were selected for validation. Additional filtering removed the measurements at which the wake of the PT or the nearby WTs affected the met mast measurements. As a change in the WT operative condition could result in an MBrM variation interfering with the load transient due to flap actuation, only flap actuation cases occurring when the WT was in almost stationary condition were selected. This selection was achieved by removing the cases where, within $10\,\mathrm{s}$ before and after the flap actuation, the pitch angle varied more than 1 deg, the rotor speed more than 0.5 rpm, and the yaw direction more than 1 rpm. Finally, a total of 150 measurements for flap activation and 135 for flap deactivation were obtained, distributed between 5 and $20\,\mathrm{ms^{-1}}$ as shown in Figure 10a.



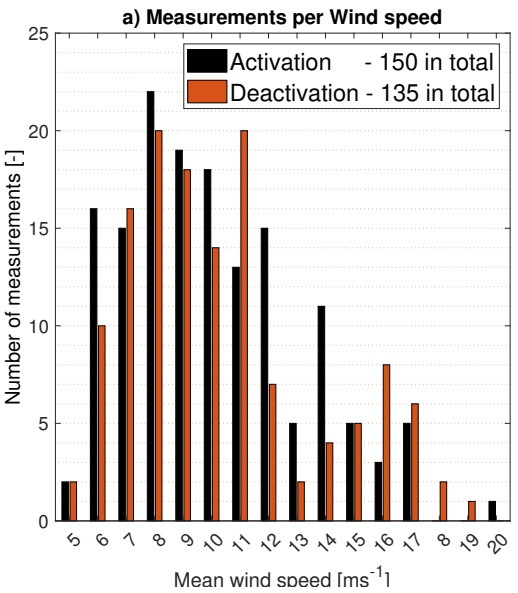

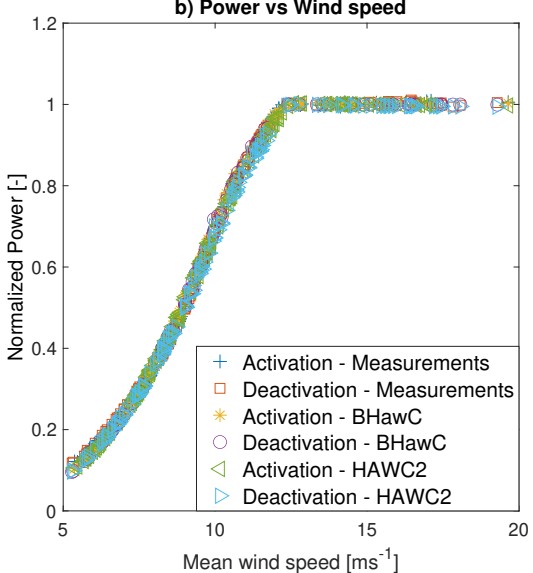

**Figure 10.** a) Number of selected measurements per wind speed for flap activation (black) and deactivation (red). b) Normalized power curve of the selected measurements and simulation (both BHawC and HAWC2) for flap activation and deactivation.

### 6.2 Models and simulations setup

The BHawC and HAWC2 models from the initial validation described in section 5 were used for this validation, including the additional $0.3\,\mathrm{s}$ deactivation delay in the flap actuator model.

To properly calculate the b2b load transient during flap actuation, the simulation setup had to match the environmental conditions at the WT at the specific time of the flap actuation. The environmental conditions were measured at the met mast, located $300\,\mathrm{m}$ in front of the PT. This distance introduced a time delay $Td$ corresponding to the time the wind needs to cover the distance $Dm$ between the met mast, where it was measured, and the WT rotor. This delay is inversely proportional to the wind speed ($Td=60\,\mathrm{s}$ for a low $5\,\mathrm{ms}^{-1}$ wind speed and $Td=15\,\mathrm{s}$ for $20\,\mathrm{ms}^{-1}$ wind) and for a discrete and constant sampling time

$\Delta T$ was calculated as

$$Td = k * \Delta T$$

where $k$ is the first time step, previous of the flap actuation time $t_0$, for which the sum of the distances traveled by the sampled wind speeds $w_{(-i)}$ covers the distance $Dm$.

$$k = \min(i \in Z^+ | \sum_{i=0}^{k} w_{(-i)} * \Delta T \geq Dm)$$

The flap actuation time $t_0$ was identified by using the flap actuation pressure gradient, as described in section 5. A $20\,\mathrm{s}$ time interval centered on the actuation time, corrected by the corresponding $Td$, was selected for each flap actuation. For each time



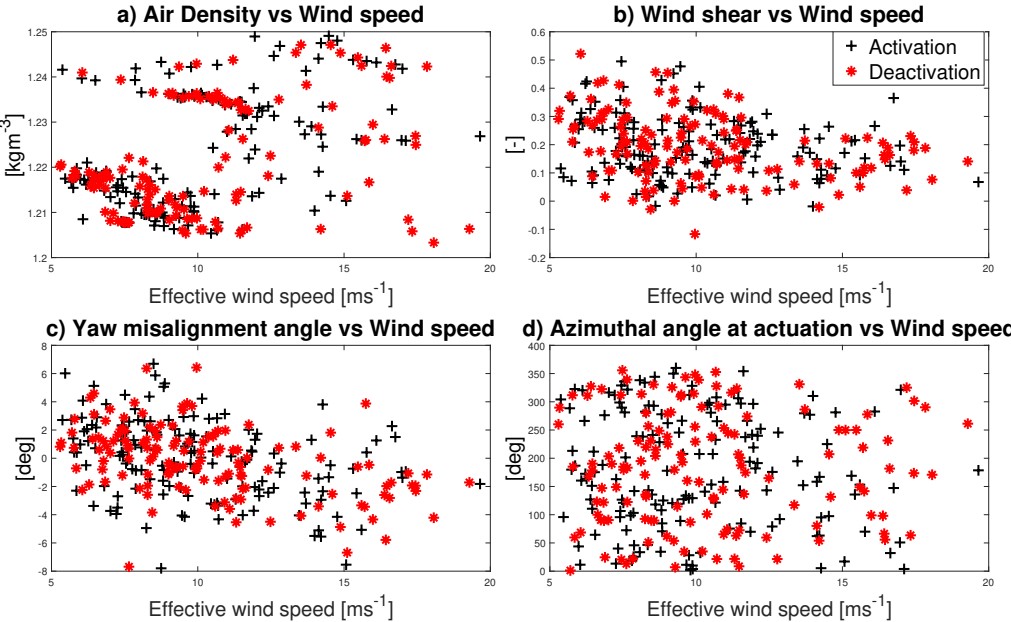

**Figure 11.** Distribution of Air density (a), Wind shear (b), Yaw misalignment angle (c) and azimuthal angle at flap actuation (d) in function of the effective wind speed used for the flap activation (black cross) and deactivation (red asterisk marker) aeroelastic simulations.

interval, the mean values of air density, wind shear, and the misalignment angle between the wind direction and the WT yaw angle were input to the aeroelastic simulations. As for the simulated wind speed, the effective wind speed value was preferred to the measured wind speed. The effective wind speed is the wind speed that makes the WT operate at the measured rotor
speed, generator power, and pitch angle. This wind speed was obtained by interpolating the measured characteristic power, pitch, and rotor speed curves at the corresponding mean values measured in the selected time interval. The MBrM transient comparison in section 5 shows the turbulence has a small impact on the average load transient due to flap actuation. This average is instead affected by the distribution of the azimuthal angle at which the flap actuation happens. These conclusions supported the decision to omit the turbulence in the current validation, avoiding the additional complexity of measuring and
modeling the equivalent turbulence. At the same time, in the simulations, the flap was actuated at the corresponding measured FA angle. Figure 11 shows the distribution of Air density (a), Wind shear (b), Yaw misalignment angle (c), and FA angle (d) in the function of the effective wind speed used in the simulations. Figure 10b shows the mean power obtained in the simulations matches the measured one for all the flap actuation cases.

## 6.3 Extended Validation results

The extended validation of the ATEF aerodynamic model relied on the comparison of the average MBrM transients. The MBrM signal was calculated and synchronized for measurements and simulations as described in section 5.





Figure 12a shows the comparison of the average MBrM transient during flap activation of the measurements (black line with asterisk marker), BHAwC simulations (blue dashed line) and HAWC2 simulations (red dash-dotted line) based on all the available data. The simulation transients are almost equivalent, with a time shift within $0.1\,\mathrm{s}$ (less than 3 time steps), bringing

the load difference within 5%. The transient pattern also differs from the transient obtained in the initial validation. The load keeps increasing and fluctuates after the flap's full activation, a behavior also shown by the measurement transient. This behavior is probably caused by the averaging of the load difference obtained from all the different wind conditions. The simulation transients are well within the error band of measurement transient, with a maximum difference below 50% of the measured std, and below 10% of the ΔMBrM due to the flap activation. Similarly to the initial validation, the simulation

transients start to increase earlier (around $0.1\,\mathrm{s}$, 5% of the actuation time of $2\,\mathrm{s}$) but with a lower slope, under-predicting the load increase in the second half of the flap activation and accumulating a delay of $0.2\,\mathrm{s}$, 10% of the actuation time) and then converging within the measured oscillating transient. As observed in the initial validation, the simulations cannot reproduce the s-shaped behavior the measurements manifest.

Figure 12b compares the average MBrM transients during flap deactivation. The simulations transients differ less than 3%, with

a time shift within $0.08\,\mathrm{s}$ (2 time steps). The simulation transients are well within the measured error band, with a maximum difference below 60% of the measured std, and below 15% of the ΔMBrM. The simulation transient decreases $0.2\,\mathrm{s}$ (4% of the $4.5\,\mathrm{s}$ deactivation time) later than the measured transient, but they quickly converge within the measurement oscillation transient.

During the actuator model tuning, the flap displays a slower activation and a faster deactivation when the WT is in normal

operation compared to the idling state. This behavior suggests that the aerodynamic loading on the blade section may influence the deflection of the flap. To investigate this hypothesis, the averaged MBrM transients were computed for wind speed ranges of $2\,\mathrm{m\,s^{-1}}$, a wind range with similar aerodynamic load values. The averaged transients are shown in Figure 13 for flap activation and Figure 13 for flap deactivation. For wind speed ranges up to $13\,\mathrm{m\,s^{-1}}$, the simulated transients closely resembled the corresponding measured transients, exhibiting differences similar to those observed in the global transients. For the range

above $13\,\mathrm{m\,s^{-1}}$, insufficient data results in irregular and oscillating transients. From the comparison of the measured averaged trends shown in figure 15a for flap activation and 15b for flap deactivation, a correlation between the wind speed and the transient shape does not emerge clearly. The measured transients all lie within a range of $0.2\,\mathrm{s}$ in both the activation and deactivation phases without a clear relation to the aerodynamic load.

## 7 Near Wake model study

In the ATEF model validations described in the previous sections, as well as in the stationary validation of the same ATEF models described in Gamberini et al. (2022), the aeroelastic codes did not model the 3D effects originated by the vorticity trailed from the edges and along the span of the flap section. HAWC2 code can account for these 3D effects via the optional Near-Wake induction model. This model, introduced by Pirrung et al. (2017), is a simplified version of the lifting line model specifically designed to examine the wake near each blade. The Near-Wake model, being fully dynamic, accounts for the



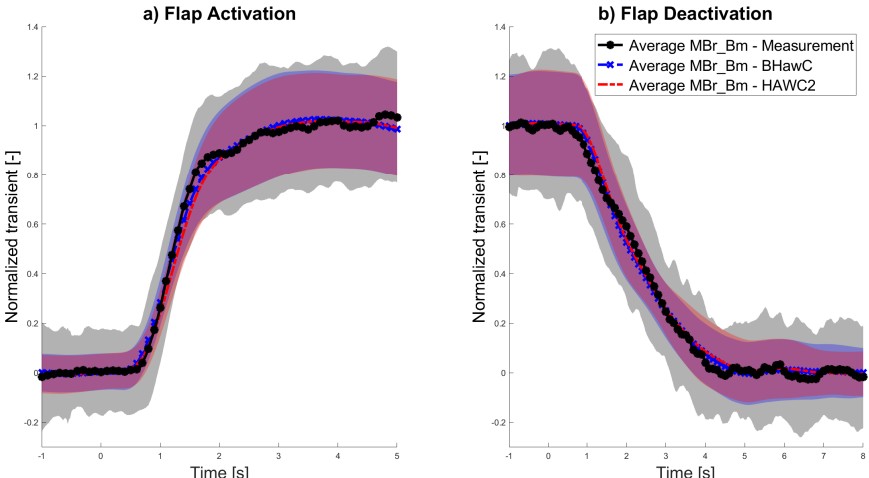

**Figure 12.** Comparison of the normalized averaged b2b blade root bending moment (MBrM) transient obtained from all BHawC (blue dashed line with x marker) and HAWC2 (red dash-dotted line) simulations and all measurements (black line with asterisk marker) during flap activation (a) and deactivation (b), including an error band of 1 std of the matching colors.

temporal evolution of the trailed vorticity. The vorticity is traced between all the aerodynamic sections on the blade, allowing the model to capture various vortices such as tip and root vortices. Furthermore, the model takes into consideration vorticity trailed at the edges of flaps and vorticity resulting from radial load fluctuations in turbulent inflow. The strength of the trailed vortex at each trailing point is determined by computing the disparity in bound vorticity between adjacent aerodynamic blade sections. This calculation incorporates an approximation of the buildup of unsteady circulation. The validation of the WT

section properties and the extended validation were performed with the Near-Wake model active in the HAWC2 simulations (H2-NW). The Near Wake model impacts the $\Delta Cl$ at the analyzed flap location, reducing the increase of Cl during flap actuation of 7% compared to the HAWC2 results without the Near Wake model (H2). The averaged Cl transients differ less than 3% of the $\Delta Cl$ during flap activation, mainly due to a time shift of almost 2 time steps, and less than 4% $\Delta Cl$ (time shift around 3 time step) during flap deactivation. The impact on the Regarding the MBrM, the H2-NW transient is almost equivalent to

the H2 transient, with a difference below 1% during flap activation and below 0.6% during flap deactivation. Additionally, the Near-Wake model impacts the $\Delta MBrM$ of the flap actuation between 2 to 2.5%.

## 8   General discussion

The validations described in sections 5 and 6 show the HAWC2 and BHawC aeroelastic models of the ATEF implemented on the PT provide almost equivalent results during the flap activation and deactivation. In the initial validation based on the

blade section measurements, the respective HAWC2 and BHawC Cl transients have a time shift below $0.04\,\mathrm{s}$, leading to a maximum difference lower than 6% of the $\Delta Cl$ occurring during flap actuation. In the extended validation, the simulated b2b



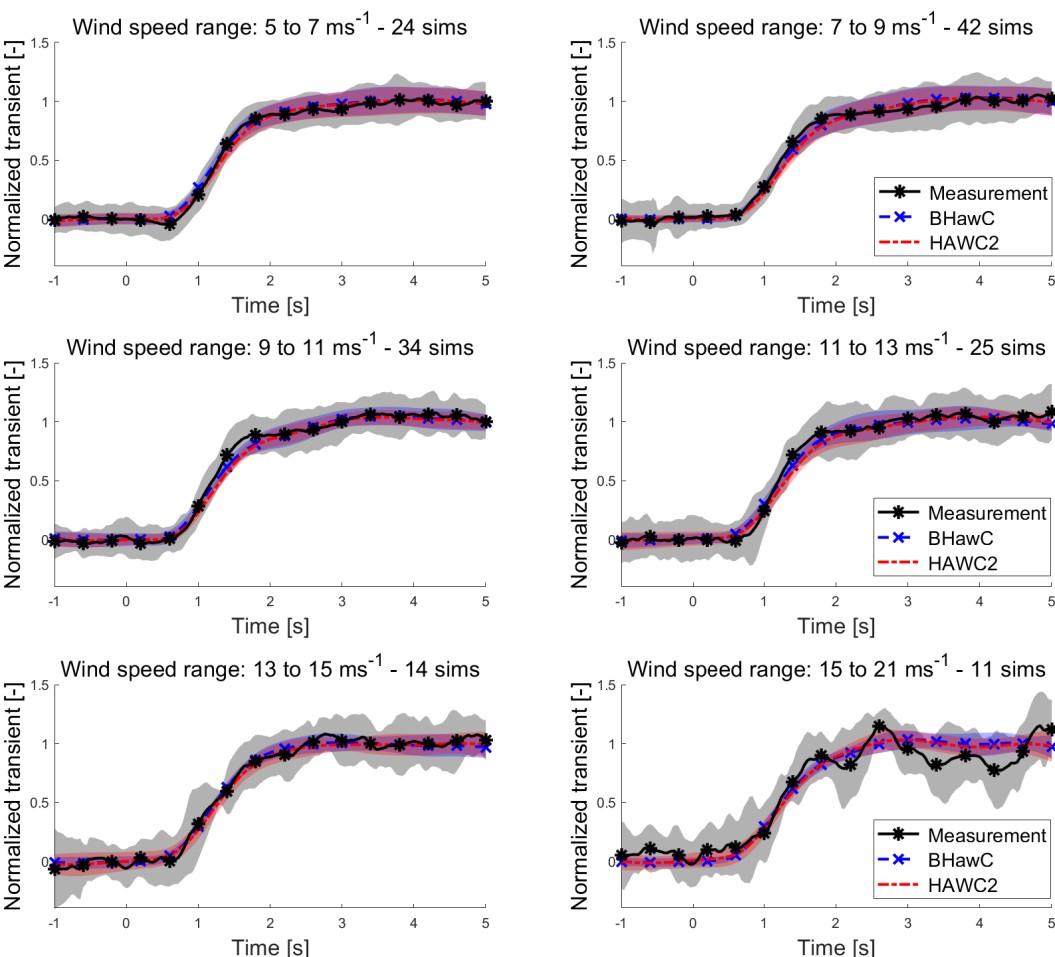

**Figure 13.** BHawC (blue dashed line with x marker), HAWC2 (red dash-dotted line), and measurements (black line with asterisk marker) averaged MBrM transients obtained for wind speed intervals of $2\,\mathrm{m\,s^{-1}}$ during flap activation. The number of data-point (sims) per wind interval is also specified

load transients differ less than 5% of the $\Delta$MBrM caused by the flap actuation, with the difference mainly originating by a maximum time shift of $0.1\,\mathrm{s}$.

The validations proved that the simulations are in good agreement with the measurements. The simulated average Cl transients are well within the error band of the measured transient, with a max difference below 8% of the $\Delta$Cl during flap activation and deactivation. The differences between the simulated and the measured average MBrM transients are below 10% of the

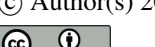

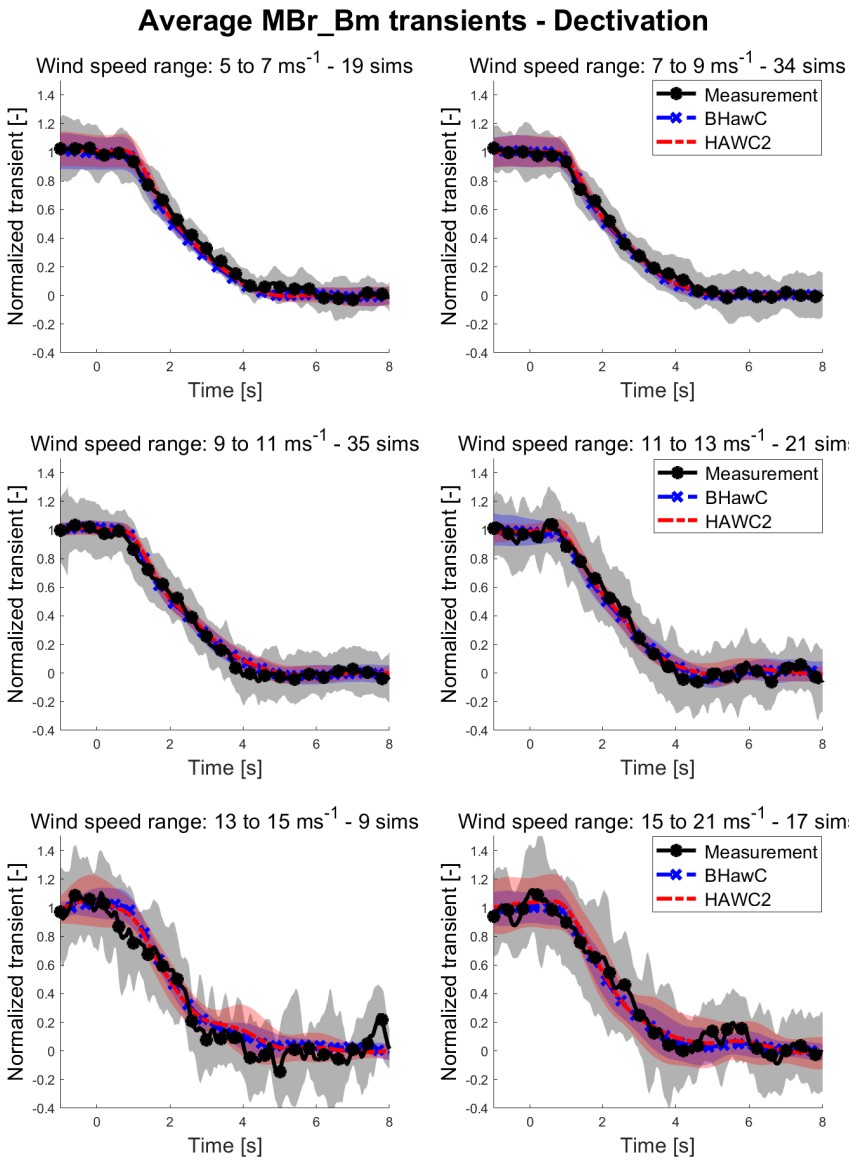

**Figure 14.** BHawC (blue dashed line with x marker), HAWC2 (red dash-dotted line), and measurements (black line with asterisk marker) averaged MBrM transients obtained for wind speed intervals of $2\,\mathrm{m\,s^{-1}}$ during flap deactivation. The number of data-point (sims) per wind interval is also specified



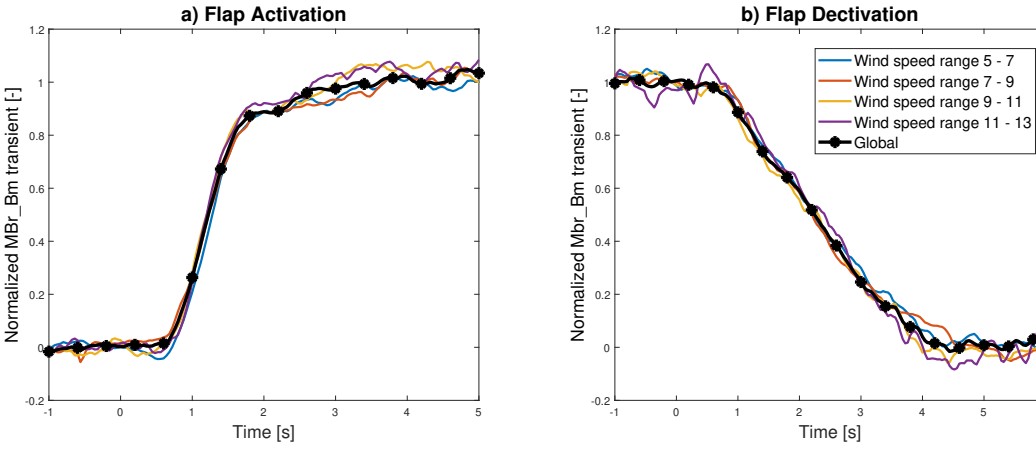

**Figure 15.** Comparison of the measured MBrM transients obtained for different wind speed intervals for flap activation (a) and flap deactivation (b)

ΔMBrM during flap activation and below 15% during flap deactivation. The ΔMBrM due to flap activation ranges between 6% (at rated wind speed) to 11% (at high or low wind speed) of the mean blade loads (as showed in Figure 4b of Gamberini et al. (2022)). Consequently, the maximum simulation transient error is below 1% of the mean blade load during activation and below 1.7% during flap deactivation. Within this error margin, the simulations cannot completely reproduce the average transients' shape, especially during the flap actuation, where the measurements rise later but steeply, reaching full activation value earlier. The main proposed causes of this difference in the transient shape are:

- Incomplete flap deflection model. The actuator model was tuned with the flap deflections measured only at one wind speed. The flap deflection transients may vary as the external conditions, like the aerodynamic forces or the rotational speed, change. As shown in Figure 15, comparing MBrM transients at different wind speed intervals does not correlate clearly with the WT operative condition. However, it still has a range of uncertainty comparable to the differences between simulations and measurements. Furthermore, the tuning flap deflections were measured only at one blade section, not ensuring the flap has a constant deflection characteristic along its whole $20\,\mathrm{m}$ length. A spanwise variation of the flap deflection could justify the steeper shape of the MBrM transient. Additional measurements of the flap deflection at different blade locations, wind speeds, and WT operative conditions can improve the deflection model, reducing the transient shape difference.

- Imperfect aerodynamic properties of the flap profiles. The Cl and Cd curves used in the aeroelastic simulations were all derived from the wind tunnel measurement of the 21% thickness profile. New wind tunnel measurements for the 18% and 24% profiles and for different relative sizes of flap and chord can verify if the flap profiles' aerodynamic properties are correct or responsible for the observed difference in the transients' shape.





- Uneven distribution of the angle at flap actuation time in the measurements. The Cl and MBrM transient are strongly affected by the azimuthal angle at which the flap actuation is initiated, as shown in section 5. In the available measurements, the FA angle of the measured flap activation and deactivation was not evenly distributed on the whole rotor, leading to a distorted averaged transient. In the o2o process, this was partly compensated by simulating the flap activation at the measured FA. Additional measurements aiming to improve the FA angle distribution at different wind speeds can potentially improve the quality of the measured transients and reduce the difference between measurement and simulation.

- Incomplete flap aerodynamic model. The transient shape difference can suggest a delay of the change in aerodynamic flow compared to the flap actuation, a delay dependent on the flap position (as it is not present during flap deactivation) that is quickly recovered during the activation. CFD simulations modeling the exact geometry of the flap deflection transient can verify this hypothesis.

Based on the experience gained in the validations presented in this paper, some recommendations are suggested to improve future validation campaigns of ATEF:

- The correct estimation of the flap deflection is crucial in the validation of the aerodynamic model of the ATEF. If continuous measurements of the flap position are not available, the flap deflection transient should be measured for several wind and operative conditions to ensure the correct tuning of the actuator model.

- Uncertainties on the aerodynamic properties of the flap profiles reduce the accuracy at which the aerodynamic model can be validated. Proper measurements of all the relevant flap profiles should be conducted. If that is not possible, the Cl and Cd impact of the flap can be derived from similar profiles with acceptable accuracy.

- The correct time synchronization of all the different measurement systems is crucial to ensure the proper time precision in measuring and validating the transient of the Cl and Load channels. Therefore, the flap actuator control (or any other channel that can be used to estimate the flap actuation time) is required.

- The azimuthal angle at which the flap is actuated strongly affects the transient of Cl and loads. The measurement campaign should aim to obtain a set of measurements with a balanced distribution of FA angles.

In section 7, the HAWC2 Near-Wake induction model (model accounting for the 3D effects due to the vorticity trailed from the edges and along the span of the flap section) does not effect the average transient of $\Delta$Cl and $\Delta$MBrM as their differ respectively less than 4% and 1% compared to the HAWC2 model without Near-Wake. The Near-Wake model impact marginally the value of $\Delta$MBrM, reducing it between 2 to 2.5%. This reduction, even if small, improve the accuracy of the model to estimate the $\Delta$MBrM as the HAWC2 model overestimate it, as shown in Gamberini et al. (2022). This result confirms the conclusions of Prospathopoulos et al. (2021), that a BEM model without 3D trailed vorticity effects overestimates the flap contribution at the flap location as well as at the blade root, but the difference of the integral loads is rather small when the flap actuation frequency is below 1P, like in the validation described in the current paper.

Finally, the azimuth-based b2b method proved to be a reliable methodology to estimate the asymmetrical loading caused by the



ATEF equipped on a single blade of the PT. This methodology can be applied in other asymmetrical rotor loading conditions, for example, in case of individual pitch, pitch error, or blade degradation.

## 9   Conclusions

Within the framework of the VIAs project, an Active Trailing Edge Flap System was installed on a $4.3\,\mathrm{MW}$ test wind turbine and underwent a field test campaign for over 1.5 years. The campaign provided the measurement data to validate the aerodynamic ATEF models of the aeroelastic engineering tools BHawC and HAWC2. The validation focused on the dynamic response of the ATEF models during flap actuation and consisted of three phases. At first, the flap actuator model was tuned to accurately reproduce the flap deflection transient during the $2\,\mathrm{s}$ activation and the $5\,\mathrm{s}$ deactivation. In the second phase, the aerodynamic flap model was tuned and validated through the lift coefficient transients measured at a blade section equipped with the flap. Finally, the aeroelastic ATEF model was validated based on the blade load transients over a three month period, from October to December 2020, with varying weather conditions. A novel approach to computing the blade load impact of the flap was introduced. This method is an azimuth-based variation of the blade-to-blade approach, and it computes the difference between azimuthal synchronized loads of adjacent blades.

The validation showed that for the tested actuator model, the two aeroelastic ATEF models provide almost identical transients during flap activation and deactivation, with the main difference caused by a relative time shift lower than $0.04\,\mathrm{s}$ (one time step) between the Cl transients and below $0.1\,\mathrm{s}$ between the load transients. The validation showed that the simulations transient of Cl and MBrM are in good agreement with the corresponding measured transients, confirming that the aeroelastic ATEF models provide a reliable and precise estimation of the impact of the flap on the wind turbine during flap actuation. In comparison with the field data, the maximum differences between the simulated and the measured Cl transient are below 8% of the $\Delta$Cl and within 0.15 s of time shift during flap activation, below 8% $\Delta$Cl and within $0.2\,\mathrm{s}$ of time shift during flap deactivation. Regarding the MBrM transients, the maximum difference is below 1% of the mean blade load during flap activation and below 1.7% during flap deactivation, with a delay within $0.2\,\mathrm{s}$ for both flap actuation cases. Additional measurements of the flap deflection at different blade locations, under wide WT operational conditions and the direct measurement of the aerodynamic properties of all flap profiles are suggested solutions to fine tune the ATEF model.

To the authors' knowledge, this is the most extensive published validation of the aeroelastic ATEF model transients in terms of wind conditions, time, and flap size. This was enabled by measuring the aerodynamic response at a blade section during flap activation/deactivation with a unique inflow and pressure belt system. Combined with the complementary validation of the static properties of the aeroelastic ATEF model, this validation increases the safety and reliability of the aeroelastic design environment for the WT equipped with active flaps and provides the basis for further exploration of the ATEF technology. Future research should aim to identify the limits of application of the ATEF models in terms, for example, of actuator performance (e.g., maximum speed or deflection) or external conditions (e.g., wind misalignment, extreme wind speed, or direction change).



*Author contributions.* A.Go conceived and planned the measurements with the support of A.Ga, T.Ba, and H.Ma. A.Go, and H.Ma performed the flyboard measurements. A.Go performed the extended validation measurements and the video recording of the flap deflection. A.Ga. performed the model tuning, the calculations, and the postprocessing, supported by T.Ba for the Cl extraction. A.Ga developed the az-b2b method from the initial b2b method developed by A.Go. A.Go and T.Ba supervised the project. All authors discussed the results. A.Ga. wrote the manuscript with input from all authors.

*Competing interests.* A.Ga. and A.Go. are hired by Siemens Gamesa Renewable Energy, company that is developing the flap technology used as reference in the paper.

*Acknowledgements.* This research was partially funded by Danmarks Innovationsfond, Case no. 9065-00243B, PhD Title: "Advanced model development and validation of Wind Turbine Active Flap system".
The Validation of Industrial Aerodynamic Active Add-ons (VIAs) project is a collaboration between Siemens Gamesa Renewable Energy, DTU Wind Energy and Rehau AS, partially funded by EUDP under journal nr. 64019-0061.
The authors thank the anonymous reviewers whose comments and suggestions helped improve and clarify this manuscript.





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
