# Peer review of "Validation of aeroelastic dynamic model of active trailing edge flap system tested on a 4.3 MW wind turbine"

_Wind Energy Science, 2023_

## Referee Comment (RC1)

Review on: Validation of aeroelastic dynamic model of Active Trailing Edge Flap system tested on a 4.3 MW wind turbine

General:

The presented work is outstanding in terms of experimental and numerical effort. However, I suggest some measures to streamline the paper. In my point of view, the motivation should be clearer, what the goals of the first and second measurement campaign are. The way it is presented now, appears rather as a technical report than a research paper.

Furthermore, I suggest only one section on experimental setup and one section on numerical setup. In the current version of the manuscript there are multiple locations where each of the aforementioned are described.

Furthermore, a clear motivation of the purpose of the flaps on a wind turbine is missing. Do the authors aim at load control employing flaps? What kind of loads? Or power control?

Furthermore, as this is a retrofit flap, do the researchers see this also as a purpose for future application? Or is an application within the design process of new turbines planned?

Moreover, a clear statement on the limitations of the flap properties is missing: What is the frequency bandwidth of the current flap? Can the flap be driven to continuous angles, or is it an on / off system?

Identified Model: As I understand it, the model was tuned at one wind speed? Is an operating point dependent model necessary? So, a gain scheduled type of model? Also, a dependency on azimuthal angle was stated. Is the azimuthal angle part of the identified model? Why did the authors not employ state-of-the-art system identification methods? Why were only step changes employed for model identification and not sinusoidal motions?

The section on the NW model is interesting but it is not clear, if this model was used for the validation in the paper. If this model was used, why is it not stated in the numerical setup section? If it wasn't used, I m not sure if the presented validation is beneficial for the current paper.

Regarding the uniqueness of the experimental setup: Did the authors encounter any differences in flap effectiveness due to erosion on the blades / actuator wear? As the authors state, this is the only known example of a flap that was tested for a long time in the field, so it would be interesting if the authors could elaborate on the degradation (if there is) of the flap / measurement results over the long measurement time.

Finally, I highly recommend a revision of English language.

Furthermore, I provide some suggestions / questions with line numbers:

Detailed Recommendations:

1) Title: Are the capital letters of 'Active Trailing Edge Flap' intentional? All other beginning letters are lower case.

2) Abstract: I recommend to avoid all abbreviations in the abstract.

3) Page1 Line2 (P1L2): There is little research on flaps for AEP increase. Is this statement necessary here? Is it a purpose of this study?

4) Abstract: I recommend present tense.

5) P1L14/15: …flap … flap… - reword

6) P2L35: activation vs. actuation

7) P2L51: as 6)

8) P3L70: specific section means specific blade section?

9) P3L92: GPS time means UTC ?

10) Generally, on Section 2: This is an impressive and unique setup. However, can the error of the measurement technique be estimated? Can the flap deflect in both directions (towards suction and pressure side)? What is the frequency bandwidth of the flap? How is the pitot tube measurement corrected for the induction of the blade? Were any changes of BRB observed on the blades that were not equipped with the flap?

11) P4Figure 1: This left photo suggests multiple flaps. Does the setup consist of one or multiple flaps? Are the gaps between the flaps due the deformation of the blade during operation?

12) P5 L111: max = maximum?

13) P5L120 controller signal = flap setpoint?

14) P5L128 what is meant by a '21% thickness flap profile'? This sentence is hard to understand.

15) P5L133 'instead' does not align with previous sentence. This section should be reworked. I miss an introductory sentence: Profile with xy% thickness, xy%thickness were equipped with ATEFs on the blade. Therefore, …

16) P6L150 how is the lift derived from the flyboard measurements? Pressure Integration of the pressure belt? Are 15 pressure taps sufficient to estimate the local lift?

17) P7165 How is the video synchronized with the valve commands? A simple encoder could not be used as a pneumatic flap is employed?

18) P7166 So the flap can be actuated in three positions? not, half and fully deflected? In my point of view, this should be stated in section 2

19) P7L172 Was a model identified each state (normal power operation and idling?)?

20) P7L175 Can you clarify what is meant by 'independent of the actual activation pressure'? The pressure to drive the flap is not considered in the model? As only a model is identified from flap setpoint to Cl ?

21) P7L182 What exactly is the 'signal of the flap controller' ? The flap setpoint?

22) P8L183 So the valve transient is ignored? Where is the pressure measured for the 'pressure channel' signal? Is the valve located in the hub or close to the flap?

23) P8 Figure 4 – 3 seconds rise time for the flap deflection seems very large. Which load cases target the authors?

24) P185L188 This model is only valid for normal power production? Is there a identified model for idling cases? How do they compare? Maybe, this could be added to the paper.

25) P8L189 'on a blade section' suggest only the pitot tube / pressure belt were used for validation. In this section the BRB is also analyzed. I suggest a change of title of this section.

26) P8L191 – How was the selected wind speed ensured during a 3 hour test in the field?

27) P8L195 – Measurements – I suggest to move this section to section 2. Thereby, one single section on measurement setup helps to streamline the paper.

28) L200 – A sampling rate of 100 Hz is fairly low. How did the authors ensure that there aren't any aliasing effects?

29) L216 This sentence is hard to understand. What is a 'full middle-pressure activation' ? I understand one cycle comprises 60s activation and 60s deactivation? How does this lead to 90s 'full activation and deactivation cycles'?

30) L222 FA angle = Flap actuation angle or azimuth angle? Please clarify the abbreviation

31) L224 So, the input to the simulation is not the met mast data? Is there a reason for this? Is it possible to input the experimental data from measurements to the simulations? How would this affect the results?

32) L225 As 29), is there a copy paste error in L216 for the times of activation / deactivation?

33) L234 The challenge of system identification with present periodic frequency components that origin from disturbances on the output signals is well known. Van Wingerden (https://ieeexplore.ieee.org/document/5497118) solved this by dividing input from actuators and input from disturbances at known frequencies. Is there a reason why the authors did not choose a system identification methodology that accounts for these different input types?

34) L252 Did the authors consider notch filters on the multiple p frequencies?

35) L260 What is a flap actuation azimuthal angle?

36) L265 Why is Cl decreasing before activation?

37) L267 What is the statement of this sentence? That the averaged CL transient is smoother than the measurements? Shouldn't this be expected?

38) L268 If there is a 5.3.1 where is 5.3.2?

39) Comparing 5c (rise time 1.5s for Cl) and 6c (rise time ~1 s for MBR) is seems that the blade root bending reacts faster than the local lift. Can the authors confirm this observation? Can it be physically explained?

40) 313 max difference of 0.1 – Which unit is this value? Is it normalized?

41) 359 What does 'with the simulation anticipating the measurement of 0.2s.' mean? Can you please clarify?

42) Figure 9 – There are multiple red and orange lines in this plot which makes this graph hard to read.

43) 384 – the yaw direction varied by 1 rpm? What does this mean?

44) Figure 10a) is the xlabel cropped?

45) Figure 12 – the y label should rather be called normalized blade root bending moment

46) L449 What is meant by 'being fully dynamic' ?

47) L459 'Regarding'

48) Section 7 – Were the previously presented results calculated with NW model or without? If this model is more accurate, why is the model without NW model employed at all?

49) L491 Which angle? Can you please clarify?

50) L515 'their' type o ?

---

## Author Comment (AC1)

**Anonymous reviewer**

**Review on**: Validation of aeroelastic dynamic model of Active Trailing Edge Flap system tested on a 4.3 MW wind turbine

**General:**

The presented work is outstanding in terms of experimental and numerical effort. However, I suggest some measures to streamline the paper. In my point of view, the motivation should be clearer, what the goals of the first and second measurement campaign are. The way it is presented now, appears rather as a technical report than a research paper.

AG: the purpose of the validation has been better described in the introduction in line 30 "*Currently, the design of commercial wind turbines heavily relies on low-fidelity aeroelastic models thoroughly validated with field measurements. Therefore, the field validation of the flap aeroelastic models is paramount for integrating the active flap into the wind turbine design. An extensive validation ensures the soundness of the simulation results, reducing the uncertainty and associated risks (and costs) that could jeopardize the introduction of active flaps in the wind turbine design.*" and line 89 "*The validation aims to enable reliable aeroelastic modeling of the load reduction strategies based on the actuation of trailing edge flaps, a fundamental milestone in the design of future wind turbines equipped with ATEF.*"

Furthermore, I suggest only one section on experimental setup and one section on numerical setup. In the current version of the manuscript there are multiple locations where each of the afore mentioned are described.

Furthermore, a clear motivation of the purpose of the flaps on a wind turbine is missing. Do the authors aim at load control employing flaps? What kind of loads? Or power control?

AG: a better description of the scope of flaps in future wind turbines is added in the introduction (line 26 "*For example, Ungurán and Kühn (2016) estimates a 10% reduction in the flapwise blade root bending moment and a 6% reduction in the tower side-side bending moment with an individual flap control strategy. These load reductions can be exploited to lower the components' cost or increase the energy production, as shown by Pettas et al. (2016) and Abbas et al. (2023), which estimated a potential reduction of the levelized cost of energy of 1.3%*" ) together with the load impact of the system installed on the prototype (line 69 "*From May 2020 to February 2021, extensive testing of the active flap was conducted with both time-fixed on-off flap actuation (shifting between two different flap positions at fixed time intervals) and 1P cyclic on-off flap actuation (cyclic activation of the flap both in-phase and counter phase with the blade azimuthal position). In time-fixed on-off actuation, the flap impacted the mean blade root flapwise bending moment between 3% and 20%, depending on the flap actuation level and wind speed. In 1P cyclic actuation, the flap showed a potential reduction of 13\% on the fatigue blade root flapwise bending moment.*")

Furthermore, as this is a retrofit flap, do the researchers see this also as a purpose for future application? Or is an application within the design process of new turbines planned?

AG: a better description of the scope of flaps in future wind turbines is added in the introduction, together with the load impact of the system installed on the prototype. Retrofitting of flap system on wind turbine will depends on SGRE commercial plans.

Moreover, a clear statement on the limitations of the flap properties is missing: What is the frequency bandwidth of the current flap? Can the flap be driven to continuous angles, or is it an on / off system? AG: the flap is an on/off system. A better description of it is added in section 2 in line 195 *"The pneumatic supply system consisted of an accumulator tank, a pump system, and a pressure valve. A remotely programmable control system regulated the target air pressure value and the pressure valve activation state (open or "to atmosphere"). The flap system allowed two actuation phases: in the activation phase, the pressure valve is open, the pressure in the hose rises to the target value, and the flap deflects to increase the local lift; in the deactivation phase, the pressure valve opens to the atmosphere, the pressure in the hose and the flap drops to zero, and the flap returns to a load neutral position. Therefore, the controller signal of the pressure valve state, named the flap control state signal in this paper, controlled the flap actuation state. Meanwhile, the target air pressure value defined the maximum flap deflection: the higher the pressure, the higher the flap deflection and the consequent local lift increase. The VIAs field campaign tested three target air pressures: low, middle, and high. The difference in angular flap deflection between the low and high actuation pressure state corresponded with approximately 10deg. However, sufficient data for the flap model validation was collected only with middle pressure."*

Identified Model: As I understand it, the model was tuned at one wind speed? Is an operating point dependent model necessary? So, a gain scheduled type of model? Also, a dependency on azimuthal angle was stated. Is the azimuthal angle part of the identified model? Why did the authors not employ state-of-the-art system identification methods? Why were only step changes employed for model identification and not sinusoidal motions?

AG: the flap actuator model was tuned with one wind speed. Figure 13 and 14 show the load responses are in good agreement for a wide ranges of wind speeds/operative conditions. Therefore, an actuator model depending on the operative condition (like a gain schedule model) does not seems necessary. However, as stated in chapter 8, additional measurements for different flap sections and at different wind and operative conditions can potentially improve the model even more.
The dependency of the azimuth angle is seen in the Cl and moment transient response, not in the flap deflection.

Only step changes were used because the flap controller system did not allow a sinusoidal actuation of the flap.  Now it is stated in line 87 *"The validation is focused only on the step flap actuation because the simple flap actuation system of the Prototype did not allow more complex controller strategies, like a sinusoidal flap actuation."*

For the actuator modelling, the followed basic approach was sufficient to characterize the system with the available data. For the Cl and moment transient response, the comparison of the azimuthal averaged response was sufficient to validate the models to a good quality. More advance techniques will be considered in future validations

The section on the NW model is interesting but it is not clear, if this model was used for the validation in the paper. If this model was used, why is it not stated in the numerical setup section? If it wasn't used, I m not sure if the presented validation is beneficial for the current paper.

AG: The main results are without the NW model. The NW model is computationally heavy, and it is available only in HAWC2. The NW model is initially omitted to study the impact of the "unsteady aerodynamic effect of the flap motion" modelled only in the ATEFlap model of HAWC2 but not in BHawC ATEF model. Chapter 7 describes the impact on CL and loads transient responses of introducing the NW model in HAWC2.

Regarding the uniqueness of the experimental setup: Did the authors encounter any differences in flap effectiveness due to erosion on the blades / actuator wear? As the authors state, this is the only known example of a flap that was tested for a long time in the field, so it would be interesting if the authors could elaborate on the degradation (if there is) of the flap / measurement results over the long measurement time.

AG: Blade erosion did not occur. The environmental conditions of the test site were rather mild and did not cause any blade erosion during the test period. The degradation of the flap system is not included in the paper because the focus is not on the mechanical characterization but rather the aeroelastic characterization. Anyways, the aerodynamic system installed on the blade performed according to expectations during a total period of approx. 4 years. Furthermore, periodic inspection was performed on the components installed in the hub without major signs of wear besides some humidity accumulation in the valves.

Finally, I highly recommend a revision of English language.

Furthermore, I provide some suggestions / questions with line numbers:

**Detailed Recommendations:**

1) Title: Are the capital letters of 'Active Trailing Edge Flap' intentional? All other beginning letters are lower case. AG: corrected

2) Abstract: I recommend to avoid all abbreviations in the abstract. AG: The use of acronyms has been reduced to the minimum in the whole manuscript.

3) Page1 Line2 (P1L2): There is little research on flaps for AEP increase. Is this statement necessary here? Is it a purpose of this study? AG: Abstract is updated and  rephrased.

4) Abstract: I recommend present tense. AG: Abstract is updated and rephrased.

5) P1L14/15: …flap … flap… - reword AG: reworded as "*The validation confirms that the studied aeroelastic models provide a reliable and precise estimation of the dynamic impact of the flap actuation on the wind turbine aerodynamics and loading, a fundamental step in the safe implementation of the active flap in the design of commercial wind turbines.*" in line 13.

6) P2L35: activation vs. actuation AG: corrected with flap "actuation"

7) P2L51: as 6) AG: corrected with "actuated"

8) P3L70: specific section means specific blade section? AG: corrected with "blade section"

9) P3L92: GPS time means UTC ? AG: No, GPS time is a time scale maintained by the atomic clocks of satellites and ground control stations of the Global Positioning System (GPS). It consists of a count of weeks and seconds of the week since 0 hours (midnight) Sunday 6 January 1980. It is now 19 seconds ahead of UTC time.

10) Generally, on Section 2: This is an impressive and unique setup. However, can the error of the measurement technique be estimated? Can the flap deflect in both directions (towards suction and pressure side)? What is the frequency bandwidth of the flap?

AG: the flap is an on/off system. A better description of it is added in section 2 in line 195 "*The pneumatic supply system consisted of an accumulator tank, a pump system, and a pressure valve. A remotely programmable control system regulated the target air pressure value and the pressure valve activation state (open or "to atmosphere"). The flap system allowed two actuation phases: in the activation phase, the pressure valve is open, the pressure in the hose rises to the target value, and the flap deflects to increase the local lift; in the deactivation phase, the pressure valve opens to the atmosphere, the pressure in the hose and the flap drops to zero, and the flap returns to a load neutral position. Therefore, the controller signal of the pressure valve state, named the flap control state signal in this paper, controlled the flap actuation state. Meanwhile, the target air pressure value defined the maximum flap deflection: the higher the pressure, the higher the flap deflection and the consequent local lift increase. The VIAs field campaign tested three target air pressures: low, middle, and high. The difference in angular flap deflection between the low and high actuation pressure state corresponded with approximately 10deg. However, sufficient data for the flap model validation was collected only with middle pressure.*"

How is the pitot tube measurement corrected for the induction of the blade? Were any changes of BRB observed on the blades that were not equipped with the flap?

AG: Detailed description of the Cl derivation process is available in "Madsen et all, Inflow and pressure measurements on a full scale turbine with a pressure belt and a five hole pitot tube, 2022"). As reported in (Gamberini et all, "Aeroelastic model validation of an Active Trailing Edge Flap System tested on a 4.3 MW wind turbine",2022), the wind turbine controller reacted to the increased torque due to flap activation with a small pitch out when it was operating around and above rated wind speed. This pitch action reduced the blade loading on all the three blades. This is another reason for the blade-to-blade moment to be used to estimate the global impact of the flap actuation.

11) P4Figure 1: This photo suggests multiple flaps. Does the setup consist of one or multiple flaps? Are the gaps between the flaps due the deformation of the blade during operation? AG: The flap system is unique. Further details of the flap structure cannot be disclosed.

12) P5 L111: max = maximum? AG: max replaced by "maximum" in the whole article

13) P5L120 controller signal = flap setpoint? AG: replaced with "flap state controller". For its definition, see comment #10

14) P5L128 what is meant by a '21% thickness flap profile'? This sentence is hard to understand. AG: replaced with "21% thickness aerodynamic airfoil of the blade equipped with flap". Also "profile" is replaced with "airfoil" in the manuscript

15) P5L133 'instead' does not align with previous sentence. This section should be reworked. I miss an introductory sentence: Profile with xy% thickness, xy%thickness were equipped with ATEFs on the blade. Therefore, … AG: Added the following lines for clarification *"The flap was installed over a longitudinal blade section with thickness ranging from 24% to 18% of the chord; therefore, the aerodynamic characteristics of the flap airfoils with thickness between 24% and 18% are needed."*

16) P6L150 how is the Cl derived from the flyboard measurements? Pressure Integration of the pressure belt? Are 15 pressure taps sufficient to estimate the local Cl? *AG: The procedure is better described in Chapter 5.1, with Madsen et al. (2022) as detailed reference*

17) P7165 How is the video synchronized with the valve commands? A simple encoder could not be used as a pneumatic flap is employed? AG: It was not possible to synchronize the video with the valve command. The synchronization among the flap pressure channel, flap deflection, Cl channel and load channel was derived during the validation process, as described from line 243 to 251.

18) P7166 So the flap can be actuated in three positions? not, half and fully deflected? In my point of view, this should be stated in section 2 AG: see comment #10

19) P7L172 Was a model identified each state (normal power operation and idling?)? AG: The data showed two different flap deflection transient response in Idling and normal operation. The flap is mainly effective during normal operation as during idling is in stalled or post stalled conditions. Therefore, the validation is focused on the normal power production and only the flap deflection data in normal production are used for the tuning of the flap model.

20) P7L175 Can you clarify what is meant by 'independent of the actual activation pressure'? The pressure to drive the flap is not considered in the model? As only a model is identified from flap setpoint to Cl ?

AG: the chapter is updated as following: *"In modeling the flap actuator, it is assumed that the activation and deactivation curves are independent of the target actuation pressure. This assumption is valid for high and middle actuation pressure scenarios covering most of the available Prototype field data. The actuator model should also include the impact of the aerodynamic loads on the flap dynamic, which varies in function of the wind speed and wind turbine operational state. Therefore, the middle and high-pressure deflection curves for normal production were selected, assuming a negligible change in the*

*impact of the aerodynamic load around the measured operative condition. The neglection of the impact of the aerodynamic loads on the total flap deflection is also based on the results from Gamberini et all. 2022, where the stationary flap properties were validated for a wide range of wind speeds.".* Furthermore in line 164 it is added the following lines *"This model directly links the controller signal of the flap state to the flap deflection, disregarding the air pressure signal. This simplification is possible because the controller system controlled only the final pressure value and the pressure valve actuation time. The variation of the air pressure inside the hose after the valve actuation depended only on the layout of the pneumatic and flap systems after the valve (for example, the length, diameter, and material of the hose and the stiffness of the flap). Therefore, the air pressure and the corresponding flap deflection were expected to have a similar transient response for each pressure valve actuation."* This states why the shape of the transient response depends only on the layout of the pneumatic and flap systems after the valve and not from the target actuation pressure.

21) P7L182 What exactly is the 'signal of the flap controller' ? The flap setpoint?  AG: see comment #10

22) P8L183 So the valve transient is ignored? Where is the pressure measured for the 'pressure channel' signal? Is the valve located in the hub or close to the flap?  AG: See comment #10

23) P8 Figure 4 – 3 seconds rise time for the flap deflection seems very large. Which load cases target the authors? AG: In the current paper the authors do not aim to any load case. The aim is to validate the aeroelastic model of the flap with the available field data. Anyway, future wind turbines will have longer blades and consequently slower rotor speeds. Therefore, also a "slow" flap will be able to address 1P loads, reducing the use and consequent wear and tear of the pitch bearing and pitch actuator.

24) P185L188 This model is only valid for normal power production? Is there a identified model for idling cases? How do they compare? Maybe, this could be added to the paper. AG: Thank you for the suggestion. The current model is valid only for power production where the flap is more impactful on loads than to idling conditions. Future study can investigate the effect of flap actuation also during idling or slow rotations.

25) P8L189 'on a blade section' suggest only the pitot tube / pressure belt were used for validation. In this section the BRB is also analyzed. I suggest a change of title of this section. AG: The title is updated

26) P8L191 – How was the selected wind speed ensured during a 3 hour test in the field? AG: Rephrased as "with an almost constant wind speed"

27) P8L195 – Measurements – I suggest to move this section to section 2. Thereby, one single section on measurement setup helps to streamline the paper. AG: Thank you for the suggestion. We evaluated that paper layout but we prefer to keep the present one with the three validation phases separated

28) L200 – A sampling rate of 100 Hz is fairly low. How did the authors ensure that there aren't any aliasing effects? AG: 100Hz is sufficient to properly detect the variation of the Cl due to flap activation (0.3 Hz in activation, 0.2Hz in deactivation) and rotor rotation (~0.2Hz for 1P)

29) L216 This sentence is hard to understand. What is a 'full middle-pressure activation' ? AG: Rephrased as *"In the selected time interval, the flap was performing on-off actuation cycles, switching between 60s*

*at middle-pressure actuation and 60s at deactivated state, completing a total of 90 cycles."* Also included a better explanation of the pressure values used for the test in chapter 2

I understand one cycle comprises 60s activation and 60s deactivation? How does this lead to 90s 'full activation and deactivation cycles'? AG: The "s" was a typo.

30) L222 FA angle = Flap actuation angle or azimuth angle? Please clarify the abbreviation. AG: FA abbreviation removed in the whole paper.

31) L224 So, the input to the simulation is not the met mast data? Is there a reason for this? Is it possible to input the experimental data from measurements to the simulations? How would this affect the results? AG: As better described later in the paper (Chapter 6.2) it is extremely challenging to derive the instantaneous wind conditions on a rotating blade section during 5 seconds of flap actuation from a punctual measurement on a met mast located 300m in front of the wind turbine. Therefore, it was used the mean wind speed of the 3 hours measurements. Measurements already characterized by an almost steady wind speed. The use of more varying wind conditions would have a limited impact on the averaged Cl and moment responses, as they are normalized and averaged for several azimuthal angles. Most probably the main effect would be the increase in the error band of the responses

32) L225 As 29), is there a copy paste error in L216 for the times of activation / deactivation? AG: See 29.

33) L234 The challenge of system identification with present periodic frequency components that origin from disturbances on the output signals is well known. Van Wingerden (https://ieeexplore.ieee.org/document/5497118) solved this by dividing input from actuators and input from disturbances at known frequencies. Is there a reason why the authors did not choose a system identification methodology that accounts for these different input types? AG: Thank you for the suggestion. We were not aware of this methodology. It will be useful in future investigations

34) L252 Did the authors consider notch filters on the multiple p frequencies? AG: The notch filter was one of the tested filtering techniques, but it didn't provide the required performances and it was discarded.

35) L260 What is a flap actuation azimuthal angle? AG: rephrased as "azimuthal angle at flap actuation"

36) L265 Why is Cl decreasing before activation? AG: The average Cl curve is not constant before and after the flap actuation for 2 main reasons: a not uniform distribution of data among the different azimuthal positions, that leads to an imperfect averaging, and turbulence, that is unlikely to be averaged out.

37) L267 What is the statement of this sentence? That the averaged CL transient is smoother than the measurements? Shouldn't this be expected?  AG: The statement is that the averaging works to remove the azimuthal impact, even if not perfectly and worst compared to the activation curve

38) L268 If there is a 5.3.1 where is 5.3.2? AG: it is a typo. Replaced by 5.4

39) Comparing 5c (rise time 1.5s for Cl) and 6c (rise time ~1 s for MBR) is seems that the blade root bending reacts faster than the local Cl. Can the authors confirm this observation? Can it be physically explained? AG & TB: As shown better in figure 9, Cl precedes the bending moment but the moments

rises faster. The behavior is observed and reproduced in the aeroelastic models.

40) 313 max difference of 0.1 – Which unit is this value? Is it normalized? AG: yes, it is normalized. Added "0.1 normalized BMD" for clarity

41) 359 What does 'with the simulation anticipating the measurement of 0.2s.' mean? Can you please clarify? AG: updated as following "The simulated BMD curve rises 0. 5s after the Cl increase, anticipating the measured curve by  0.2s."

42) Figure 9 – There are multiple red and orange lines in this plot which makes this graph hard to read. AG: color changed

43) 384 – the yaw direction varied by 1 rpm? What does this mean? AG: corrected with "1 deg"

44) Figure 10a) is the xlabel cropped? AG: Figure updated

45) Figure 12 – the y label should rather be called normalized blade root bending moment AG: Figure updated

46) L449 What is meant by 'being fully dynamic' ? AG: Removed

47) L459 'Regarding'. AG: comment not clear

48) Section 7 – Were the previously presented results calculated with NW model or without? If this model is more accurate, why is the model without NW model employed at all? AG: The previous results are without the NW model. The NW model is computationally heavy, and it is available only in HAWC2. The NW model is initially omitted to study the impact of the "unsteady aerodynamic effect of the flap motion" modelled only in the ATEFlap model of HAWC2 but not in BHawC ATEF model.  Chapter 7 describes the impact on CL and loads transient responses of introducing the NW model in HAWC2.

49) L491 Which angle? Can you please clarify? AG: corrected with "azimuthal angle"

50) L515 'their' type o ? A: corrected with "they"

---

## Author Comment (AC2)

The article describes a validation of two aeroservoelastic models using a unique dataset obtained in the field from a 4.3MW wind turbine where one blade is equipped with a trailing edge flap. The work is novel and very relevant. The article is fairly well written and I recommend the publication of the article. Before publication, I think the authors could improve the article further by following some suggestions:

1) Although slightly out of scope, it would be nice and relevant to read about the intended use of such flap system. Both the abstract and the introduction say that the ATEF system is promising. Why is that? Promising for what?

AG: example of potential benefits added in line 26 as follow "*For example, Ungurán and Kühn (2016) estimates a 10% reduction in the flapwise blade root bending moment and a 6% reduction in the tower side-side bending moment with an individual flap control strategy. These load reductions can be exploited to lower the components' cost or increase the energy production, as shown by Pettas et al. (2016) and Abbas et al. (2023), which estimated a potential reduction of the levelized cost of energy of 1.3%.*"

2) The authors use acronyms heavily. In my current job we have a communications department that oversees our manuscripts, and I've learnt that the use of acronyms should be minimized to improve readability AG: The use of acronyms has been significantly reduced. MBrM replaced by BMD.

3) The document relies heavily on the word "transient", which I found fairly confusing. What does a "maximum difference for the blade-to-blade MBrM transients below 1%" mean? Isn't enough to say "maximum difference for the blade-to-blade MBrM below 1%". We know MBrM varies in time/azimuth/wind speed. This is only one of the many uses of the word transient that I found confusing. AG: The use of transient word is improved and clarified in the whole document

4) you've split the validation in three steps: first the flap deflections, second the lift coefficients, third the full aeroelastic model. Although this is said multiple times, it doesn't always come out clearly. Maybe a scheme could help, or a clear numbered list in the intro? AG: The 3 phases has been numbered in the introduction (line 93). The Step number has been added also to the chapter name and in the figure 3

5) HAWC2 and BHAWC are similar models and indeed the results match very well between the two. Would it help to only report results from the former? AG: the purpose of the paper is also to show how close the results of the 2 codes are for the specific application, therefore results from both codes are included. It is clarified in the manuscript in line 85

6) Several paragraphs are very dense and not always clear. Maybe some schemes would help quick readers glance through the document. AG: Figure 3 now shows the flap model component involved in the 3 validation steps. Several improvements to the manuscript have been done to improve readability.

And some additional minor suggestions:

- Line 1: Why is "Wind Turbine" capitalized? AG: removed as the abstract is updated

- Line 25: it would be interesting to read more about the "potential benefits". AG: See comment #1

- Line 107: parenthesis seems missing wrapping Bergami and Gaunaa, 2012. AG: parenthesis added

- Line 112: typo, initial vs indicial AG: Indicial is correct

- Lines 120-124: I don't follow this paragraph. Can you please rephrase it? AG: Rephrased as "*This simplification is possible because the controller system controlled only the final pressure value and the pressure valve actuation time. The variation of the air pressure inside the hose after the valve actuation depended only on the layout of the pneumatic and flap systems after the valve (for example, the length, diameter, and material of the hose and the stiffness of the flap). Therefore, the air pressure and the corresponding flap deflection were expected to have a similar transient response for each pressure valve actuation.*"

- Line 131: mainly 4 million. Why mainly? AG: most of the measurements were run at 4 milions. Few at 3.5 or 3 milions. Rephrased with "The measurements, most of them run at a Reynolds number of 4 million," in line 178

- Line 139: lift and drag coefficients don't depend on the chord. Why were they adjusted to the chord? And how?
TB: The correction accounts for the flap chord percentage, so scaling the wind tunnel data (variation of the coefficients) for the actual percentage of the flap in full scale. The text has been re-phrased for clarity.

- Line 165-170: I find this paragraph somewhat hard to follow. AG: The paragraph has been rewritten as "*The BHawC flap model directly provides the instantaneous*

*stationary 2D aerodynamic properties to the global wind turbine model. Instead, the HAWC2 ATEFlap model computes already the unsteady effects due to flow separation and the vorticity shed into the wake, providing the instantaneous dynamic aerodynamic properties to the global wind turbine model."*

- Figure 4: I was surprised to see a transient of 3 seconds. Isn't the pitch actuator faster that that? Again a little out of scope, but the value of a "slow" ATEF becomes harder to justify. AG: future wind turbines will have longer blades and consequently slower rotor speeds. Therefore also a "slow" flap will be able to address 1P loads, reducing the use and consequent wear and tear of the pitch bearing and pitch actuator. Nevertheless, the VIAs project had the aim to develop the flap technology, building the knowledge for future and faster flap systems.

- Line 191: how can you target one wind speed experimentally? AG: Rephrased as "with an almost constant wind speed"

- Line 217: 60s+60s=120s (not 90)? AG: The "s" was a typo. Rephrased as *"completing a total of 90 cycles"*

- Line 235: this issue should be discussed further up or the previous paragraphs are confusing AG: The explanation is expanded here and removed from the previous chapters

- Line 237: if you average across azimuth, shouldn't the azimuth-variation be gone entirely? AG: the azimuth-variation is entirely gone only if the data is evenly distributed among symmetric azimuthal angles (minimum every 90 deg). Unfortunately, this is hardly achievable with measurements.

- Lines 248: I would put this paragraph first, and then the analysis. AG: Thank you for the suggestion. AG: We prefer to keep the current order: Azimuth variation of Cl, Simulations, Measurements, Comparison between measurements and simulations

- Figure 6: the legend covers the label: AG Figure updated

- Figures 12/13/14: include the grey band in the legend : AG Adding the error bands in the legend will make it significantly bigger and reduce the readability of the figure. The error band are anyway mentioned in the captions

- Line 480: I see a good match in Figure 15, what am I missing? AG: There is not a clear correlation between the BMD transient curves and the wind speed interval. However, the small differences between the averaged curves from different wind speed intervals have a magnitude comparable the validation error margin. This suggests the variation

of the external conditions can partly justify the differences observed between simulations and measurements.

- Line 494: example of excessive use of acronyms, what's o2o AG: acronym for one-to-one. Removed as it is rarely used in the paper

- Line 515: typo, affect AG: corrected

- Line 517: typo, improves AG: corrected

- Line 518: typo, overestimates AG: corrected

- Line 537: isn't a DT of 0.04 s excessively large? AG: 0.04 is the sampling time of the prototype data acquisition system. The simulation time step is actually 0.02s. The error has been corrected in the whole manuscript

- Line 565: thank you for the acknowledgment, even before reviews were in! To be seen if they improve the paper ;) – AG: You are welcome